# Dynamic tracking of native precursors in adult mice

**Suying Liu[1,2], Sarah E Adams[1,2], Haotian Zheng[3], Juliana Ehnot[1,2], Seul K Jung[1,2], Greer Jeffrey[1,2], Theresa Menna[1,2], Louise Purton[4,5], Hongzhe Lee[3], Peter Kurre[1,2]***

[1]Comprehensive Bone Marrow Failure Center, Children's Hospital of Philadelphia, Philadelphia, United States; [2]Perelman School of Medicine, University of Pennsylvania, Philadelphia, United States; [3]Department of Biostatistics, Epidemiology and Informatics, University of Pennsylvania, Philadelphia, United States; [4]Stem Cell Regulation Unit, St. Vincent's Institute of Medical Research, Fitzroy, Australia; [5]Department of Medicine, The University of Melbourne, Parkville, Australia

## eLife Assessment

This **important** study by Liu and colleagues uses lineage tracing of hematopoietic stem and progenitor cells in situ to infer the clonal dynamics of adult hematopoiesis. The authors apply a new mathematical analysis framework enabling a wider range of clonal estimation and the revised study (1) provides evidence of polyclonal adult hematopoiesis, (2) provides insights on clonal dynamics during fetal liver hematopoiesis, and (3) reveals unexpectedly high polyclonality in a mouse model of bone marrow failure (Fanconi anemia), arguing against the prevalent views of clonal attrition in this context. The evidence in this extensively revised and improved study is **compelling**, with methods, data and analyses more rigorous than the current state-of-the-art, which will be of broad interest not only to stem cell and developmental biologists working on hematopoiesis, but also to researchers working on other systems.

*For correspondence:
KURREP@email.chop.edu

Competing interest: The authors declare that no competing interests exist.

**Abstract** Hematopoietic dysfunction has been associated with a reduction in the number of active precursors. However, precursor quantification at homeostasis and under diseased conditions is constrained by the scarcity of available methods. To address this issue, we optimized a method for quantifying a wide range of hematopoietic precursors. Assuming the random induction of a stable label in precursors following a binomial distribution, estimates depend on the inverse correlation between precursor numbers and the variance of precursor labeling among independent samples. Experimentally validated to cover the full dynamic range of hematopoietic precursors in mice ($1-10^5$), we utilized this approach to demonstrate that thousands of precursors, which emerge after modest expansion during fetal-to-adult transition, contribute to native and perturbed hematopoiesis. We further estimated the number of precursors in a mouse model of Fanconi Anemia, showcasing how repopulation deficits can be classified as autologous (cell proliferation) and non-autologous (lack of precursor). Our results support an accessible and reliable approach for precursor quantification, emphasizing the contemporary perspective that native hematopoiesis is highly polyclonal.

## Introduction

Continuous self-renewal and differentiation of hematopoietic stem and progenitor cells (HSPCs) is fundamental to blood production. Even though rare cases exist where a single HSPC clone supports hematopoiesis, the HSPC population contributing to homeostatic hematopoiesis is usually highly

polyclonal (*Sun et al., 2014*; *Yamamoto et al., 2013*). For example, the number of HSPCs actively participating in white blood cell production was estimated to range from 20,000 to 200,000 in humans (*Lee-Six et al., 2018*; *Mitchell et al., 2022*). Conversely, a decline in the number of active hematopoietic precursors has been linked to hematopoietic dysfunction. For example, in humans older than 70 years of age, hematopoietic dysfunction coincides with an abrupt reduction in the HSPCs population actively contributing to blood production (*Mitchell et al., 2022*).

These observations underscore the association between a low number of active hematopoietic precursors and hematological function. Yet, few methods are suitable for quantifying active hematopoietic precursors in a native environment (*Lee-Six et al., 2018*; *Mitchell et al., 2022*; *Breivik, 1971*; *Campbell et al., 2023*; *Wallis et al., 1975*; *Szilvassy et al., 1990*). In mice, methods utilizing in situ barcodes often suffer from barcode homoplasy, preventing total precursor number estimation (*Bowling et al., 2020*; *Pei et al., 2017*). In humans, estimations based on somatic mutations and computational modeling carry a high degree of uncertainty (*Lee-Six et al., 2018*; *Mitchell et al., 2022*). The absence of precise precursor quantification methods in their natural environment hampers the study of precursor numbers across different conditions and their potential use as predictive functional markers.

In mice, the quantification of developmental hematopoietic precursors has been achieved using animals engineered with a *Confetti* cassette (*Ganuza et al., 2017*). This cassette can randomly recombine and express one of four fluorescence proteins (FPs, being RFP, CFP, YFP, or GFP) upon Cre induction, resulting in variability in Confetti expression pattern in mice that inversely correlates with precursor numbers (*Figure 1A*; *Ganuza et al., 2017*). While suitable for developmental hematopoiesis, this method has limited linear range (50–2500), impeding its potential application to adult hematopoiesis or clonally restricted hematopoiesis. To measure hematopoietic precursors in various conditions, expansion of the detection range is desired.

Interestingly, the random induction of Confetti colors in HSPCs bears similarity with X-chromosome inactivation (XCI). In XCI, one of the X chromosomes is randomly inactivated in precursor cells, a process that adheres to a binomial distribution. Formula derivation from binomial distribution implies that a higher XCI variance among individuals correlates with a smaller number of precursors at the time of inactivation (*Figure 1B*). As XCI is faithfully maintained in the progeny, this correlation has been used to estimate precursor numbers from mature blood cells in human and mouse, leading to the discovery of the first mutation (Tet2) that contributes to clonal hematopoiesis (*Wallis et al., 1975*; *Buescher et al., 1985*; *Berger and Sturm, 1996*; *Busque et al., 2012*). Nonetheless, XCI occurs exclusively in females, limiting its applicability in males. Additionally, XCI takes place during early development when few precursors are present, resulting in low sensitivity in adulthood (*Gandini et al., 1968*).

Building on insights from the *Confetti* mice and XCI studies, we investigated whether random induction of Confetti FPs in precursor cells can be similarly modeled by a binomial distribution, whereby the variance of FPs inversely correlates with precursor numbers. Based on this relationship, we asked if we could broaden the measurable range of precursor number, which would allow us to expand the existing correlation range.

Examining the premises of binomial distribution and setting some assumptions, we report that the random induction of FPs among a group of mice or cells can be modeled by a binomial distribution. Experimental validation establishes a broadened linear range, covering the full spectrum of precursor numbers ($1–10^5$) and overcoming the prior range limit. We leverage this correlation to probe the number of hematopoietic precursors at homeostasis, post-myeloablation, during developmental expansion, and in a mouse model of inherited bone marrow (BM) failure.

## Results
### Binomial distribution underlies the inverse linear correlation between FP% variance and the precursor numbers

To broaden the limited correlation range, we aimed to elucidate the mathematical relationship between variance and precursor numbers (*Ganuza et al., 2017*). Inspired by XCI, we asked how the stochastic induction of Confetti FPs in precursors may satisfy the premises of binomial distribution (*Micklem et al., 1987*; *Harrison et al., 1987*). We found that when we assume the number

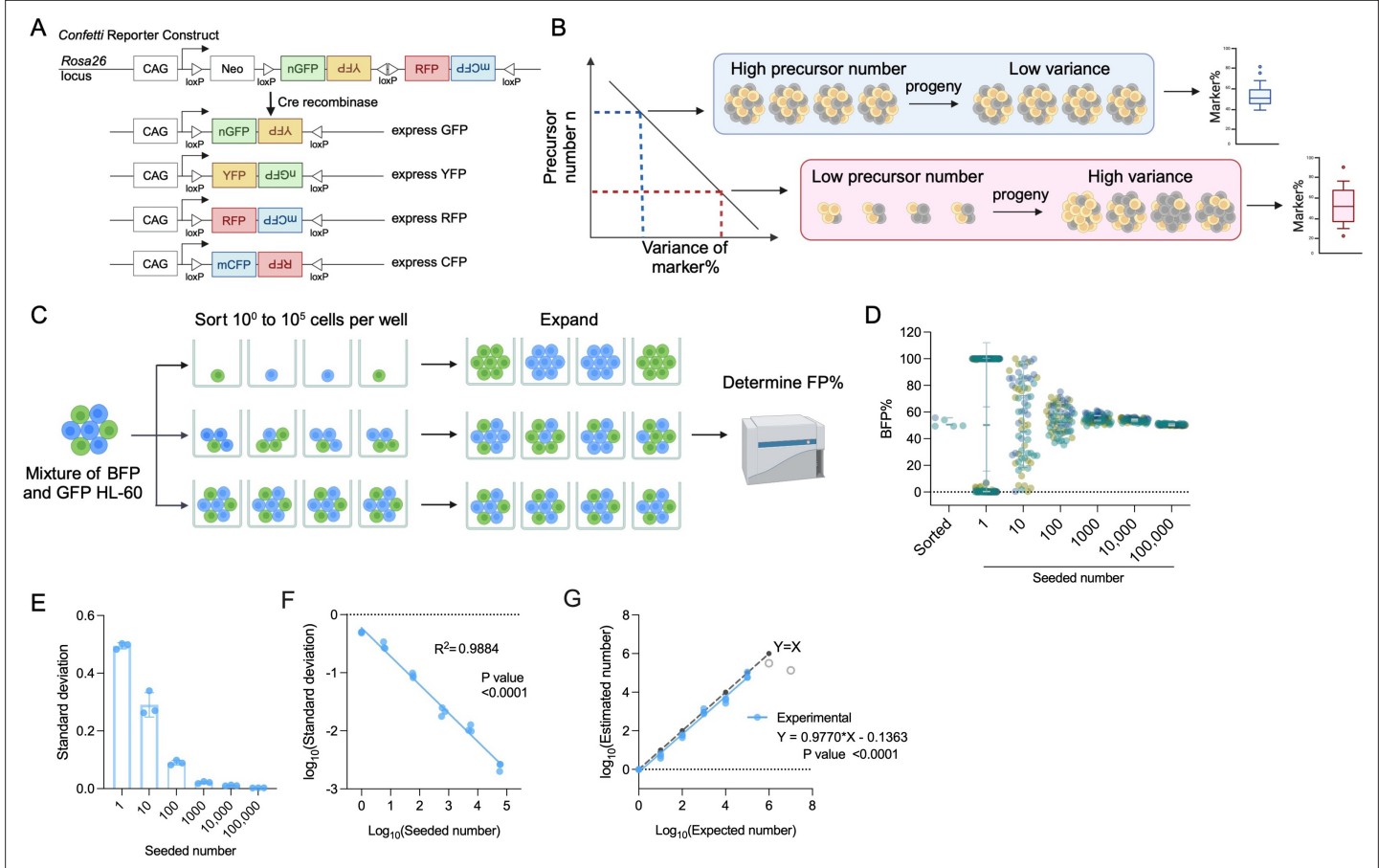

**Figure 1.** The principle of inverse correlation between variance of FP% and precursor numbers. (**A**) Schematic of the *Confetti* cassette at the mouse *Rosa26* locus. Sequences of four fluorescence protein are interspaced by loxP sites. Upon Cre-expression, the *Confetti* cassette will recombine and express one of the four fluorescence proteins. (**B**) The inverse relationship between variance of markers and precursor numbers. The distribution of FP% is determined in the progeny to estimate the number of precursors. (**C**) Workflow to validate the correlation formula between variance of FP% and precursor numbers by a two-color cell system. (**D**) Blue fluorescent protein (BFP) frequencies in wells seeded with 1–100,000 HL-60. Each dot represents a well. Two replicates were shown. Each replicate consists of at least 20 wells per seeded number. Error bars represent mean ± SD. (**E**) The standard deviation of BFP% in wells seeded with various numbers of HL-60. Error bars represent mean ± SD. $N = 3$ replicates. (**F**) The correlation between seeded numbers and standard deviation of BFP%. $N = 3$ replicates. (**G**) The correlation between seeded numbers and the numbers estimated with standard deviation of BFP% and *equation 2*. $N = 3$ replicates except for seeded numbers $10^6$ and $10^7$ (gray circles). For $10^6$ and $10^7$, $N = 1$ replicates. For (**A–C**), images were created with BioRender.com/f96e500.

The online version of this article includes the following source data and figure supplement(s) for figure 1:

**Source data 1.** Original data for plotting *Figure 1*.

**Figure supplement 1.** Establishing the correlation between variance of fluorescence protein (FP) and precursor numbers.

**Figure supplement 1—source data 1.** Data: original data for plotting *Figure 1—figure supplement 1*.

of precursors remain the same for the condition tested, then induction of FPs meets the required premises: (1) the number of precursors, denoted as *n*, remains constant within a group of mice or cells; (2) each observation (e.g., each precursor) is independent; (3) each observation yields one of two outcomes (e.g., it signifies the presence or absence of RFP in a precursor); (4) the probability of success, represented as *p* (e.g., the likelihood of a precursor expressing RFP upon Confetti induction), remains consistent across all observations.

Should the induction of a FP adhere to a binomial distribution, precursor numbers can be estimated using the following equation:

$$\lg\left(\hat{n}\right) = \lg\left[\frac{\left(1 - \widehat{FP_{mean}\%}\right)}{\widehat{FP_{mean}\%}}\right] - 2 \times \lg\left(\widehat{CV_{FP}}\right) \tag{1}$$

where $n$ signifies the estimated number of precursors, $FP_{mean}\%$ represents the estimated probability of a precursor being one given FP (e.g., $RFP_{mean}\%$), and $CV_{FP}$ denotes the estimated coefficient of variance (CV) of one given FP (e.g., $CV_{RFP}$). CV is the standard deviation divided by the mean, utilized by the empirical correlation formula.

The equation proposes a direct linear relationship between the precursor number and the CVs of individual FPs, supporting the empirical correlation formula (*Ganuza et al., 2017*). Indeed, by analyzing a published dataset providing precursor numbers and the corresponding distribution of FPs, we found that the correlation between $n$ and individual CVs of FP exhibits superior fit (>0.93), higher than the empirical correlation equation based on all three CVs (0.75) (*Figure 1—figure supplement 1A*; *Ganuza et al., 2017*). Moreover, according to *equation 1*, the $y$-intercept of correlation between precursor number and CV was influenced by $FP_{mean}\%$, while the slope was the same (−2) regardless of $FP_{mean}\%$. Given the unequal value of $RFP_{mean}\%$, $CFP_{mean}\%$, and $YFP_{mean}\%$ in the published dataset, the correlation equations show significantly distinct $y$-intercepts ($p < 0.0001$) (*Figure 1—figure supplement 1A, B*). In contrast, the slopes of these correlation equations were similar ($p = 0.49$, with all 95% confidence intervals encompassing −2, the theoretical value) (*Figure 1—figure supplement 1A*).

In conclusion, the FP induction in precursor cells can be modeled by a binomial distribution with the assumption that precursor numbers are constant among a group of mice. This sets the mathematical basis for the inverse linear correlation between variance of FP% and the number of precursors.

## A broad range of precursor numbers correlates with variance of FP%

While the correlation formula derived from binomial distribution does not impose any range limitation, experimental errors may confound the measurement of variance. We next aimed to experimentally confirm this correlation. To streamline the calculation, we used standard deviation instead of CV to compute variance. The correlation between standard deviation and precursor numbers is expressed through the following equation:

$$\lg\left(\hat{n}\right) = \lg\left[\widehat{FP_{mean}\%} \times \left(1 - \widehat{FP_{mean}\%}\right)\right] - 2 \times \lg\left(\widehat{\sigma_{FP}}\right) \tag{2}$$

where $\widehat{\sigma_{FP}}$ denotes the estimated standard deviation of a given FP% (e.g., $\widehat{\sigma_{RFP}}$).

Given this equation, $\lg(\widehat{\sigma_{FP}})$ exhibits a linear and inverse correlation with $\lg\left(\hat{n}\right)$. Indeed, $\lg(\widehat{\sigma_{RFP}})$, $\lg(\widehat{\sigma_{CFP}})$, and $\lg(\widehat{\sigma_{YFP}})$ exhibited a linear and inverse correlation with $\lg\left(\hat{n}\right)$ in the published data (*Figure 1—figure supplement 1C*; *Ganuza et al., 2017*).

To simplify the validation, we used a two-color cell model HL-60 bearing one of the two FPs (BFP and GFP, wherein GFP represents non-BFP) (*Figure 1C*). Although the *Confetti* cassette recombination can generate one of four colors, our estimation is based on a single given FP (e.g., a precursor expresses RFP or not), making this simplification justifiable.

We first proved that neither BFP nor GFP HL-60 cells had a competitive growth advantage over the other, ensuring that HL-60 progeny mirrored the seeding population (*Figure 1—figure supplement 1D*). We next sorted one to $10^7$ cells into individual wells and allowed them to proliferate for at least three generations before assessing BFP% at the end of the culture (*Figure 1C*). We focus on cell range up to $10^5$ cells as $1.4 \times 10^5$ non-hematopoietic stem cell (HSC) LSK and $5.2 \times 10^3$ active HSCs were previously estimated in an adult female mouse, which corresponds to the highest possible precursor numbers in mice (*Cosgrove et al., 2021*). We considered the survival rate of cells in wells receiving a single HL-60 cell (53–60%), to ensure the accuracy of the number of cells seeded.

Consistent with the proposed correlation, as seeded numbers increase, BFP% standard deviation decreases (*Figure 1D, E*). The variance of BFP% reveals an inverse correlation with the seeding numbers ($R^2 = 0.996$) (*Figure 1F*). After normalizing for cell survival rates, the calculated numbers closely align with the expected numbers from one to $10^6$ cells (*Figure 1G*). We had anticipated that measurement of small variance in high-precursor numbers may be confounded by experimental errors. Indeed, at cell numbers larger than $10^6$, the standard deviation of BFP% and the expected number reached a plateau (*Figure 1G*, gray circles). To ensure accurate measurement without confounding errors, we suggest $10^5$ to be the upper limit for experimental measurement.

## Experimental practices for accurate precursor number measurement

It is crucial to ensure the accuracy of variance measured because variance of FP% is solely used for estimation. Based on the data from the two-color HL-60 model, we conclude that there are at least three experimental practices required for accurate cell estimates: (1) exclusion of outliers; (2) sufficient flow cytometry recorded events; (3) sufficient sample size per group. Both outliers and insufficient recorded events inaccurately inflate sample variance, leading to underestimation of precursor numbers (*Figure 1—figure supplement 1E, F*). We found that the minimum recorded events increase in tandem with the seeded number, in contrast to the reported 500-event threshold (*Figure 1—figure supplement 1F*; *Ganuza et al., 2017*).

While having a small sample size per biological replicate is possible, the variability of estimates among replicates would be high (*Figure 1—figure supplement 1G*). Practically, achieving a large sample size (>20) is cumbersome in mice. To determine a feasible sample size for relatively accurate estimates, we resampled datapoints to calculate the variability reduction as sample size increases (*Figure 1—figure supplement 1H*). Our analyses reveal that five samples per group are sufficient to substantially minimize error, whereas an additional increase in sample size leads to marginal error reduction. We therefore use at least five mice per biological replicates for our precursor estimations.

## Inducible HSC-SCL-CreER$^T$-targeted Confetti labeling in the active blood precursors at the adult stage

To label hematopoietic precursors and quantify their number in vivo, we considered the choice of Cre mouse line to be crossed with the *Confetti* mouse. Historically, hematopoietic precursors are thought to be HSCs capable of long-term repopulation. However, recent studies indicate that multipotent progenitors (MPPs) also contribute alongside HSCs in maintaining blood production (*Fanti et al., 2023*; *Patel et al., 2022*; *Schoedel et al., 2016*; *Solomon et al., 2024*). Hence, both HSCs and MPPs should be labeled.

Among Cre mouse lines capable of labeling HSCs and MPPs simultaneously at the adult stage (such as Rosa26$^{CreERT2}$, Mx1-Cre, and HSC-SCL-CreER$^T$), we chose HSC-SCL-CreER$^T$ because of its preference for labeling HSPCs (immunophenotypically defined as Lin$^-$Sca-1$^+$cKit$^+$ (LSK)) (*Göthert et al., 2005*). To validate the specificity of HSC-SCL-CreER$^T$, we generated mice possessing a single *Confetti* allele and homozygous HSC-SCL-CreER$^T$ alleles. We then examine Confetti expression 1 day after a 2-day tamoxifen administration. Consistent with prior reports of HSC-SCL-CreER$^T$ activity, we observed Confetti expression in the LSK population and T cells (*Figure 2A*, *Figure 2—figure supplement 1A–E*). Unexpectedly, Confetti expression was additionally detected in NK cells, CD41$^+$ cells, non-inflammatory monocytes (SSC-A$^{low}$Ly6C$^-$Ly6G$^-$CD11b$^+$), common myeloid progenitors (CMP, Lin$^-$Sca-1$^-$cKit$^+$CD16/32$^{-/low}$CD34$^+$), and granulocyte–monocyte progenitors (GMP, Lin$^-$Sca-1$^-$cKit$^+$CD16/32$^+$CD34$^+$) (*Schoedel et al., 2016*; *Figure 2—figure supplement 1D–G*). Considering the relatively shorter lifespan of these cells compared to HSC/MPP, the labeling in these cells should minimally impact precursor calculations, especially with a chase period post-labeling. While the Confetti induction in the relatively long-lived T cells precludes estimation based on mature T cells, we conclude that HSC-SCL-CreER$^T$ remains a practical choice for labeling active hematopoietic precursors.

The stability of Confetti labeling after induction is required for binomial distribution. Therefore, the absence of background Cre activity after Confetti induction is critical. We determine the background Cre activity in *HSC-SCL-CreER$^T$/Confetti* animals by two approaches: (1) detecting Confetti expression in non-induced animals; (2) identifying cells co-expressing two Confetti FPs months after tamoxifen induction due to cassette 'flipping' from background Cre activity. We observe no Confetti-expressing cells without tamoxifen treatment in mice up to 50 weeks old and minimal co-expression of two Confetti FPs in induced *HSC-SCL-CreER$^T$/Confetti* animals (*Figure 2—figure supplement 2A, B*). Conversely, *Vav-Cre/Confetti* animals constitutively express Cre, resulting in ~15% cells co-expressing two Confetti colors in peripheral blood (PB) T cells (*Figure 2—figure supplement 2C, D*). Both lines of evidence support minimal Cre background activity in HSC-SCL-CreER$^T$ and its use as a Cre driver for precursor number calculation.

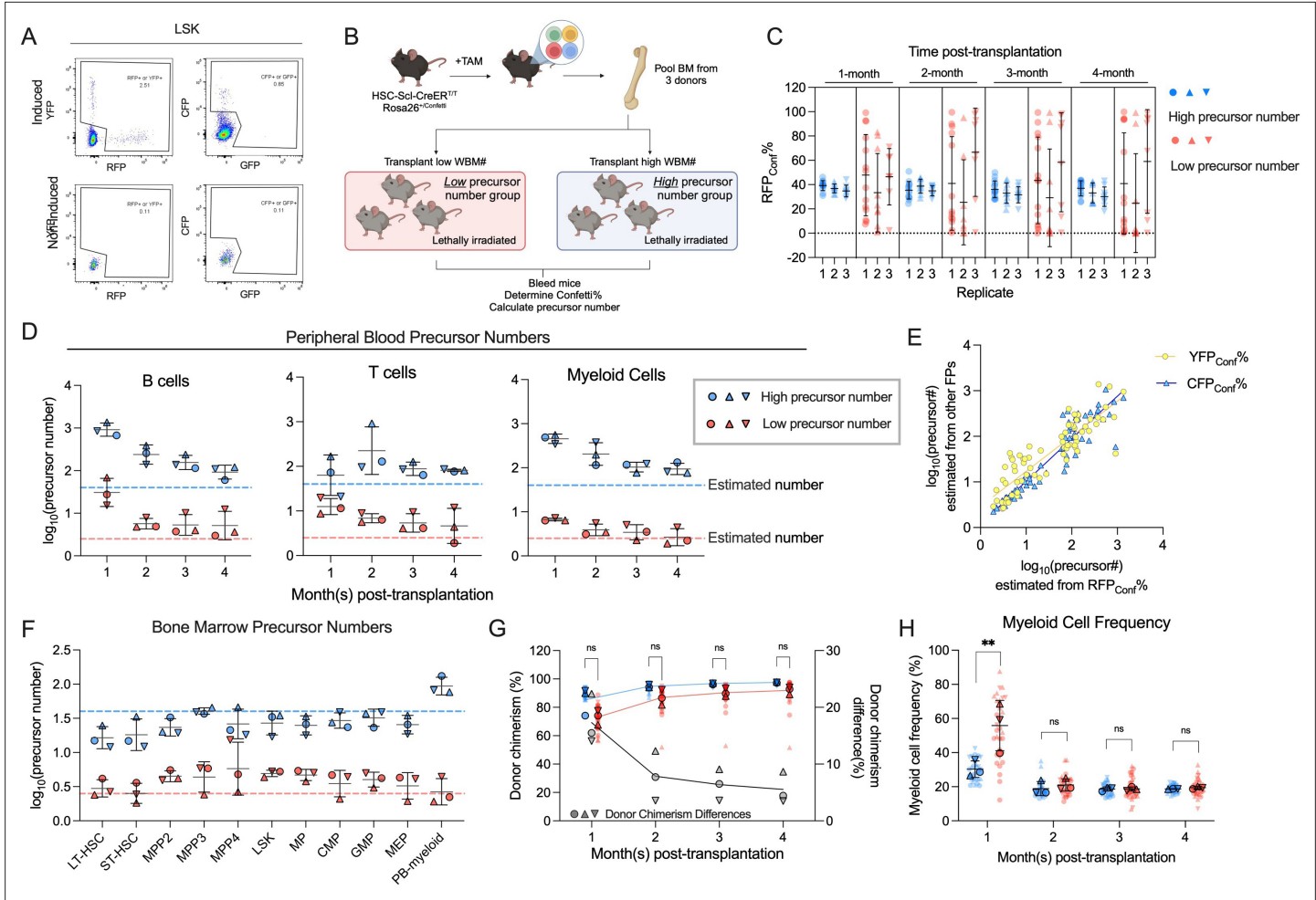

**Figure 2.** Transplantation of defined number of precursors. (**A**) Representative flow plots showing Confetti induction by HSC-SCL-CreER[T] in the LSK population. (**B**) Schematic for transplantation of defined number of precursors. Briefly, $4 \times 10^6$ Confetti-induced WBM for 'high-precursor number' groups, or $0.25 \times 10^6$ WBM for 'low-precursor number' groups were transplanted into lethally irradiated recipient mice. Image was created with BioRender.com/f96e500. (**C**) $RFP_{Conf}\%$ distribution in peripheral blood (PB) myeloid cells. Each dot represents one animal. Recipient sample size = 7–14 mice per replicate, $N$ = 3 replicates per group. Error bars represent mean ± SD. (**D**) The precursor numbers of B cells, T cells, and myeloid cells in recipient mice. Dash lines mark the level of historically estimated transplantable clone numbers. Each dot represents a precursor number calculated from multiple mice. Error bars represent mean ± SD. (**E**) The correlation of precursor numbers calculated from $RFP_{Conf}\%$, $YFP_{Conf}\%$, or $CFP_{Conf}\%$. Each dot represents a precursor number calculated from multiple mice. (**F**) The precursor number of bone marrow (BM) MP and hematopoietic stem and progenitor cell (HSPC) in recipient mice. Dash lines represent historically estimated transplantable clone numbers. Each dot represents a precursor number calculated from multiple mice. Error bars represent mean ± SD. (**G**) The donor chimerism of recipient mice, and the CD45.2[+] chimerism differences between high- and low-precursor number groups. Each solid dot represents one animal. Each outlined dot represents the average value of a replicate containing multiple animals. Recipient sample size = 7–14 mice per replicate, $N$ = 3 replicates for each precursor number group. Two-way ANOVA for paired samples was performed. (**H**) The frequency of myeloid cells in the PB of recipients. Each dot represents one animal. Recipient sample size = 7–14 mice per replicate, $N$ = 3 replicates for each precursor number group. Two-way ANOVA for paired samples was performed. ns, non-significant, *$p < 0.05$, **$p < 0.001$.

The online version of this article includes the following source data and figure supplement(s) for figure 2:

**Source data 1.** Original data for plotting *Figure 2*.

**Figure supplement 1.** The induction specificity of HSC-SCL-CreER[T].

**Figure supplement 1—source data 1.** Data: original data for plotting *Figure 2—figure supplement 1*.

**Figure supplement 2.** Analysis of HSC-SCL-CreER[T] background Cre activity.

**Figure supplement 2—source data 1.** Data: original data for plotting *Figure 2—figure supplement 2*.

**Figure supplement 3.** Variance of FP% inversely correlates with precursor numbers in vivo.

**Figure supplement 3—source data 1.** Original data for plotting *Figure 2—figure supplement 3*.

*Figure 2 continued on next page*

*Figure 2 continued*

**Figure supplement 4.** Bone marrow (BM) analysis for high- and low-precursor number groups.

**Figure supplement 4—source data 1.** Original data for plotting *Figure 2—figure supplement 4*.

## The variance of FP% inversely correlates with precursor numbers in vivo

To ascertain the inverse correlation between variance of FP% and the number of hematopoietic precursors in vivo, we first investigate the variance of FP% in mice hosting various numbers of hematopoietic precursors. We generate these mice by non-competitively transplanting $0.25 \times 10^6$ or $4 \times 10^6$ induced CD45.2$^+$ *HSC-SCL-CreER$^T$/Confetti* mouse BM cells into CD45.1$^+$ recipient mice (*Figure 2B*). Given the constant frequency of precursors in the donor BM ($1/10^5$) and the varying doses of donor BM cells, these mice received approximately 2.5 (low-precursor number) or 40 (high-precursor number) precursors (*Micklem et al., 1987*; *Harrison et al., 1988*; *Harrison et al., 1990*; *Harrison et al., 1993*). While similar experiments transplanting defined numbers of precursors have been reported, the estimation for precursor numbers lower than 50 has not been explored.

Due to the scarcity of hematopoietic precursors in the BM, the precursor number seeded in recipients followed Poisson distribution instead of being constant, violating the premise (1) of binomial distribution. Nonetheless, our simulations show that the seeding number variation among recipient mice is relatively small, therefore the variance resulting from precursor number differences still dominates and inversely correlates with precursor numbers. A minor problem is that the estimated numbers are not at one-to-one ratio to expected numbers at low-precursor number range (<10, see Methods) (*Figure 2—figure supplement 3A*).

As cells are not 100% labeled with Confetti, for clarity, we use 'FP$_{PB}$%' or 'FP$_{BM}$%' to represent the frequency of a given FP in the PB or BM, and use 'FP$_{Conf}$%' to represent the frequency of a given FP in the Confetti$^+$ population. For total precursor number calculations, we normalize the number calculated from the Confetti-labeled population to Confetti labeling efficiency.

As expected, we observe higher RFP$_{Conf}$% variance across all PB cell types in mice belonging to the 'low-precursor number' groups (*Figure 2C*, *Figure 2—figure supplement 3B, C*). Since variance of any FP following a binomial distribution inversely correlates with precursor numbers, we also find higher CFP$_{Conf}$% and YFP$_{Conf}$% variance in the 'low-precursor number' groups (*Figure 2—figure supplement 3D, E*). Following the same principle, we observe higher RFP$_{PB}$%, CFP$_{PB}$%, YFP$_{PB}$%, and Confetti% variance in 'low-precursor number' mice (*Figure 2—figure supplement 3F*).

Applying *equation 2*, we find that the precursor numbers of B cells and myeloid cells are noticeably higher in the first 2 months than at 4 months post-transplantation, suggesting transient progenitor contributions at early timepoints (*Figure 2D*). At 4 months post-transplantation, the estimated precursor numbers align with the expected values (in myeloid cells, 94 for 'high-precursor number' groups, three for 'low-precursor number' groups, *Figure 2D*). Again, for any FP following binomial distribution, variance of FP% inversely correlates with precursor numbers and thus can be leveraged to calculate precursor numbers (Methods). Since different FPs measure the same population of precursors, we expect the estimations from different FPs to be similar. Indeed, precursor numbers derived from RFP$_{Conf}$% highly correlate with those estimated from YFP$_{Conf}$% or CFP$_{Conf}$% ($R^2 = 0.885$ for CFP% in Confetti$^+$, $R^2 = 0.769$ for YFP% in Confetti$^+$), as well as those from RFP$_{PB}$%, YFP$_{PB}$%, and CFP$_{PB}$% (*Figure 2E*, *Figure 2—figure supplement 3G*).

Consistent with PB data, higher variance of RFP$_{Conf}$% is also observed in the BM of 'low-precursor number' mice (*Figure 2—figure supplement 4A*). The precursor numbers estimated from various BM HSPC subpopulations align well with each other and are consistent within the same group (*Figure 2F*). However, in 'high-precursor number' groups, estimates from BM HSPCs are lower than those derived from PB myeloid cells (27 for BM HSPC, 94 for PB myeloid cells, *Figure 2F*). This discrepancy may reflect uneven seeding of precursors to the BM throughout the body after transplantation, and the fact that we only sample a part of the BM (femur, tibia, and pelvis) (*Li et al., 2023*). In summary, we validate the inverse correlation between variance of FP% and precursor numbers in vivo. Moreover, the quantification of precursor numbers is feasible even for precursor numbers outside the empirical correlation range.

## Cell frequency measurements fail to reflect the differences in precursor numbers

Given that transplantation studies are performed non-competitively and the recipients are lethally irradiated, we expect to see minimal differences in donor chimerism between two groups, despite drastic differences in donor precursor numbers. Indeed, although we observe substantially lower donor chimerism in mice belonging to 'low-precursor number' groups during the first 2 months post-transplantation, the chimerism differences are very small by 4 months post-transplantation (5.5 ± 2.8%, *Figure 2G*). The initial differences in donor precursor numbers do not affect PB cell frequencies, except for the first month post-transplantation, when higher PB myeloid frequencies are observed in the 'low-precursor number' groups (*Figure 2H*). This suggests myeloid cell production is enhanced when very few precursors are available after irradiation-mediated injury (*Figure 2C*).

In the BM, nucleated cell counts, donor chimerism, and HSPC frequencies are mostly similar regardless of donor precursor numbers, suggesting that even extremely low numbers of hematopoietic precursors can still effectively repopulate a non-competitive environment (*Figure 2—figure supplement 4B–D*). Therefore, in cases where hematopoietic precursors expand or decline without competition, stem cell frequencies are less informative to study precursor activity. Measuring precursor numbers is more meaningful as low number of active precursors may be constrained by compensatory proliferation.

## Thousands of hematopoietic precursors contribute to native hematopoiesis

While the empirical formula, which measures 50–2500 precursors, supports quantification of precursors labeled at fetal stages, it may not be applicable to native hematopoiesis, when the total precursor number may exceed 2500. Having validated the feasibility for measuring precursor numbers in vivo by *Confetti* animals, we next sought to investigate the number of active precursors contributing to native hematopoiesis. In *HSC-SCL-CreER$^T$/Confetti* animals, we opt to label a smaller portion of HSPCs with a 2-day treatment, despite the potential to label up to 60% HSPCs with Confetti (CFP + YFP + RFP) by 14-day tamoxifen treatment (*Figure 3A*, *Figure 3—figure supplement 1A*). This precautionary measure aims to mitigate potential toxicity arising from prolonged tamoxifen treatment.

Unexpectedly, we observed a successive decrease of CFP$_{Conf}$% and CFP$_{PB}$% in some animals, which distort the distribution of CFP$_{Conf}$% (*Figure 3—figure supplement 1B, C*). A similar decline is not observed in RFP$_{Conf}$% or YFP$_{Conf}$%, nor in RFP$_{RFP+YFP}$% (calculated by RFP$_{PB}$%/(RFP$_{PB}$% + YFP$_{PB}$%)) (*Figure 3—figure supplement 1D, E*). A similar loss of CFP$_{Conf}$% is neither observed in animals with Confetti induction during fetal development, suggesting a potential immune response to CFP in adult-induced animals (*Figure 3—figure supplement 1F*). To circumvent this caveat, we decided to calculate precursor numbers solely based on the variance of RFP$_{RFP+YFP}$%. Since only ~2% of myeloid cells are labeled with CFP, we reason that the decline of CFP in some animals is unlikely to affect the calculation of total precursor numbers based on other FPs, such as RFP$_{RFP+YFP}$%. Moreover, in transplantation studies, precursor numbers calculated with variance of RFP$_{RFP+YFP}$% linearly correlate with those calculated using variance of other FPs (*Figure 2—figure supplement 3G*).

To further ensure accuracy, we focus on PB myeloid cells, as (1) Confetti labeling of T cells result from direct induction by HSC-SCL-CreER$^T$, but not differentiation from HSPCs (*Figure 2—figure supplement 1D*); (2) Confetti labeling (RFP and YFP) of B cells does not plateau at 7 months post-induction (*Figure 3B*); (3) myeloid cells are of shorter lifespan. In myeloid cells, we focus on precursor numbers after 4 months post-induction, when their labeling reaches a plateau (*Figure 3B*). The stability of average RFP$_{RFP+YFP}$% suggests an equal contribution of RFP$^+$ and YFP$^+$ cells to the PB, a prerequisite for estimating precursor numbers from FP% distributions in the progenies (the myeloid cells) (*Figure 3C*). Like those in PB, the average RFP$_{RFP+YFP}$% in BM HSPC subpopulations is also stable, supporting non-biased differentiation of RFP$^+$ and YFP$^+$ precursor cells (*Figure 3C*).

Fitting data to *equation 2*, we estimate an average of 2667 precursors contributing to native myelopoiesis (*Figure 3D*, average of numbers at 5–7 months post-induction). This number closely aligns with the clone number estimated by transposon-based barcodes statistic in granulocytes (831/30% = 2770) (*Sun et al., 2014*).

The average precursor number calculated from PB myeloid cells at the time of BM analysis (1958) matches those calculated from BM myeloid progenitors (MP, Lin$^-$Sca-1$^-$cKit$^+$) and HSPCs (1773 and

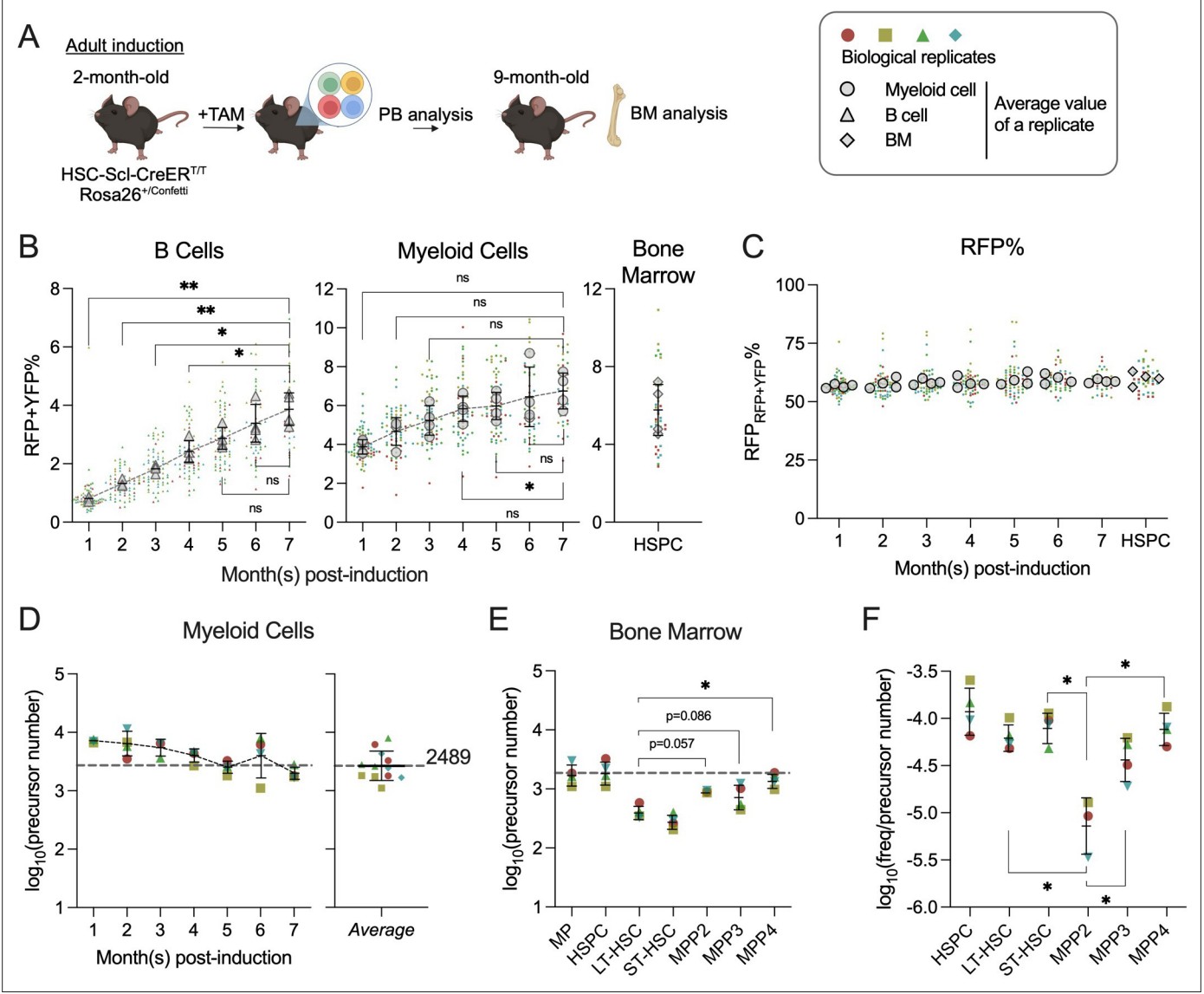

**Figure 3.** Quantification of active hematopoietic precursor at steady state. (**A**) Experiment schematic for Confetti induction in adult-induced animals. Image was created with BioRender.com/f96e500. (**B**) Confetti labeling in B cells, myeloid cells, and bone marrow (BM) hematopoietic stem and progenitor cells (HSPCs). Statistics for comparisons between labeling of month 7 and other months were shown. One-way ANOVA for paired samples was performed. (**C**) RFP$_{Conf}$% in peripheral blood (PB) myeloid cells and BM HSPCs. Statistics for comparisons between labeling of month 7 and other months were shown. One-way ANOVA for paired samples was performed. (**D**) The number of myeloid precursors and the average precursor number from month 5 to 7. (**E**) The number of BM precursors. Paired permutation was performed. (**F**) The frequency-to-clone ratio of BM HSPCs. Paired permutation test was performed. Each solid dot represents one animal. Each outlined dot represents the average value of a replicate containing multiple animals. Error bars represent mean ± SD. PB sample size = 14–22 mice per replicate; BM sample size = 8–10 mice per replicate; $N$ = 4 replicates. ns, non-significant, *p < 0.05, **p < 0.001.

The online version of this article includes the following source data and figure supplement(s) for figure 3:

**Source data 1.** Original data for plotting *Figure 3*.

**Figure supplement 1.** Analysis of fluorescence protein (FP) induction in adult-induced animals.

**Figure supplement 1—source data 1.** Original data for plotting *Figure 3—figure supplement 1*.

1917), but it is fivefold higher than that of LT-HSC (*Figure 3E*). Although LT-HSCs are thought to sustain steady-state hematopoiesis, recent studies suggest that ST-HSC and MPPs persist long term as primary contributors to adult hematopoiesis (*Sun et al., 2014*; *Fanti et al., 2023*; *Schoedel et al., 2016*; *Busch et al., 2015*). The observed discrepancy between the number of precursors contributing

to LT-HSC and those contributing to MPPs indicates that at least some MPPs are not replaced by progenitors directly differentiated from LT-HSC at 7 months post-induction, affirming the persistence of MPPs (*Figure 3E*). Furthermore, the fact that PB myeloid and BM MP precursor numbers are closer to those of MPPs than LT-HSC confirms active and long-term (at least 7 months) contributions from MPPs to steady-state myelopoiesis.

Of note, the quantity of precursor does not necessarily correlate with the frequency of cell type in the BM. For example, the cell frequency of MPP2 (Lineage⁻cKit⁺Sca-1⁺Flk2⁻CD48⁺CD150⁺) in BM is very low, but MPP2 comprises a modest number of precursors, making its frequency-to-precursor-number ratio the lowest among the HSPC subtypes (*Figure 3F* and S6G). The frequency-to-precursor-number ratio reflects how well precursors of a particular type proliferate to expand their absolute cell count. The low frequency-to-precursor-number ratio of MPP2 suggests that it expands more poorly than other HSPC subtypes.

In summary, we detect thousands of hematopoietic precursors contributing to adult hematopoiesis. At the time of BM analysis, the number of PB myeloid precursors is comparable to those observed in BM MP and HSPCs.

## Precursor numbers determined by FP% variance confirm reduced clonality of progenitors after myeloablation

Myeloablation through 5-fluorouracil (5-FU) treatment depletes most actively cycling cells, forcing quiescent stem cells to proliferate (*Wilson et al., 2008*). Previous studies have indicated a significant reduction in the number of clones detected within the BM c-Kit⁺ population (consisting of MP and HSPC) following a single dose of 5-FU treatment (*Bowling et al., 2020*). However, questions arise regarding whether this is an artifact stemming from potential under-sampling of the highly expanded c-Kit⁺ population after 5-FU treatment by single-cell sequencing. Given the quantitative estimation of a wide range of precursor numbers through *equation 2*, we aimed to investigate whether precursor numbers within progenitor populations indeed reduce 10 days post-5-FU treatment (*Figure 4A*).

To determine the precursor changes post-5-FU treatment, we use the animals described in *Figure 3A* as untreated (UT) benchmark cohort, which were collected at the same age. At 10 days post injection, the efficacy of 5-FU treatment is validated by lower PB myeloid cell frequency post-treatment and higher frequencies of BM progenitors compared to UT animals (*Figure 4—figure supplement 1A, B*). The high-progenitor frequencies in the BM results from the depletion of cycling cells in the BM and the enhanced proliferation of HSPCs following 5-FU treatment.

Compared to UT animals, the precursor numbers of MP (including CMP, GMP, and MEP) and HSPC significantly decrease, confirming a reduction in clonality in the c-Kit⁺ population (*Figure 4B*; *Bowling et al., 2020*). In contrast, the precursor numbers of LT-HSC do not show a decreasing trend. While transplantation studies support unchanged clonality of primitive stem cells after a single dose of 5-FU, a similar investigation in a native environment has not been conducted (*Lerner and Harrison, 1990*). Our findings support the notion that native LT-HSC clonality remains unaltered following one-dose 5-FU treatment. Together, myeloablation treatment reinforces how the dynamics of precursor numbers can be tracked through Confetti pattern variations.

## Modest developmental expansion of active lifelong precursors

HSCs are thought to undergo substantial expansion in the fetal liver during fetal development (*Morrison et al., 1995*; *Ema and Nakauchi, 2000*; *Rybtsov et al., 2016*). However, a recent study employing the empirical formula challenges this notion by quantifying endogenous lifelong hemato-poietic precursors labeled at various developmental stages (*Ganuza et al., 2022*). It revealed limited expansion of hematopoietic precursors during the fetal liver stage (from E10 to E15, 1.8- to 2.7-fold) as well as a gradual and moderate increase from the fetal liver to the post-natal stage (2.4- to 10-fold) (*Ganuza et al., 2022*). Although intriguing, most post-natal measurements in this study fell outside the empirical linear range, leaving the genuine degree of post-natal precursor expansion uncertain. As our formula quantitatively assesses a wide range of precursor numbers, we set out to directly compare the numbers of precursor labeled at various development stages (E11.5 and E14.5, by one dose of tamoxifen; adult-stage, benchmark shown in in *Figure 3A*, all analyzed at the same age) (*Figure 4C*).

For accuracy, we first examine the dynamics of Confetti labeling. In E14.5-induced animals, the Confetti labeling of T cells almost double from 1 to 4 months of age (4.7% at 1 month, 9.3% at

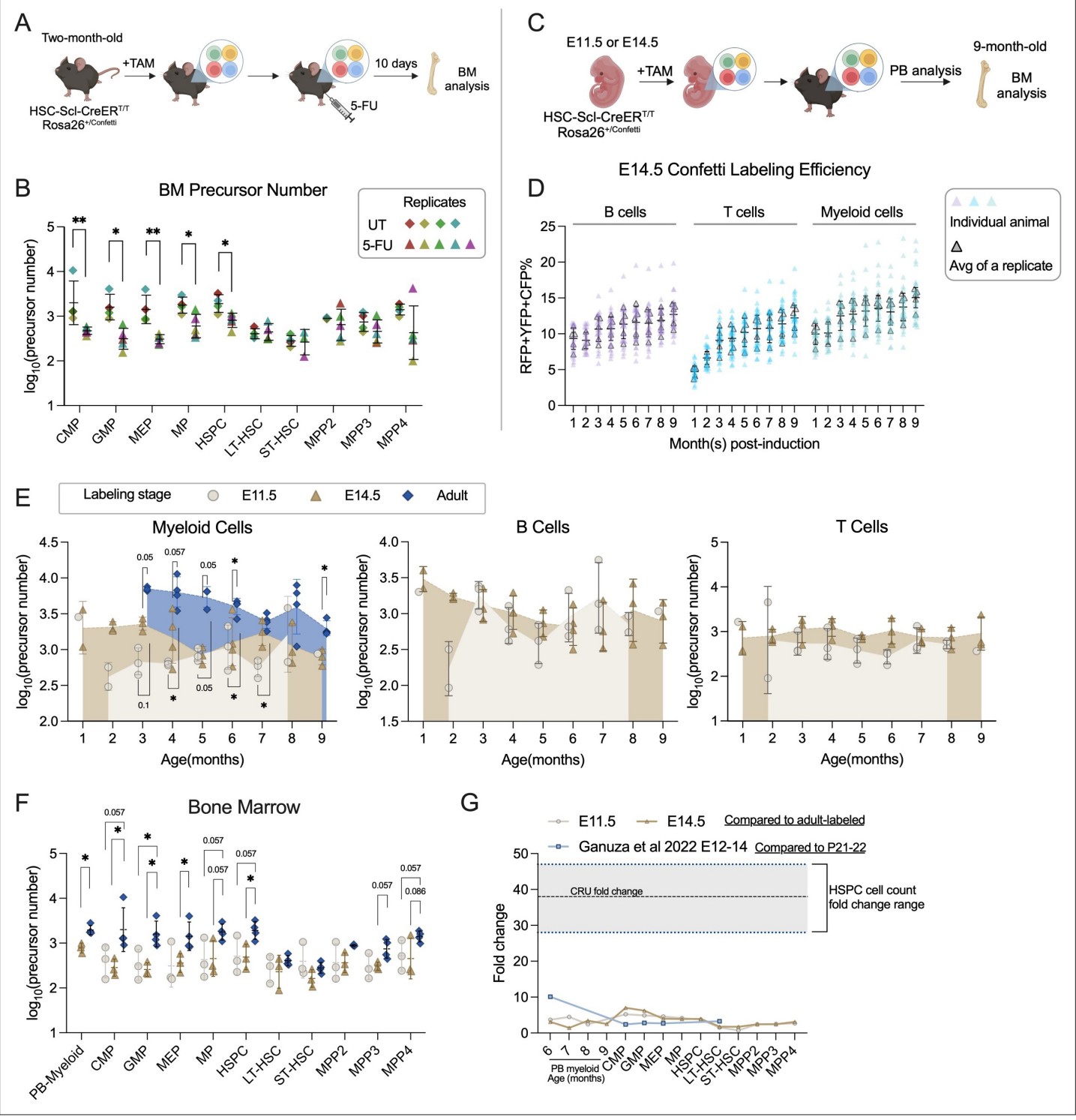

**Figure 4.** Number of precursors post-5-FU treatment and along developmental ontogeny. (**A**) Experiment schematic for one-dose 5-FU treatment. Image was created with BioRender.com/f96e500 . (**B**) The number of bone marrow (BM) precursors post-5FU treatment. Permutation test was performed. For UT, sample size = 8–10 mice per replicate; *N* = 4 replicates; for 5-FU, sample size = 6–8 mice per replicate; *N* = 5 replicates. (**C**) Experiment schematic for Confetti induction at fetal stages. Image was created with BioRender.com/f96e500. (**D**) The Confetti labeling of B cells, T cells, and myeloid cells. Each dot represents one animal. Each outlined dot represents the average value of a replicate containing multiple animals. Sample size = 4–11 mice per replicate; *N* = 5 replicates. (**E**) The number of precursors in peripheral blood (PB) myeloid cells, B cells, and T cells. Permutation test was performed. For E14.5 and adult-induction, *N* = 4 replicates; for E11.5, *n* = 3 replicates. (**F**) The number of BM precursors. Each dot represents one precursor number. Permutation test was performed. For E14.5 and adult-induction, *N* = 4 replicates; for E11.5, *N* = 3 replicates. (**G**) The relative fold

*Figure 4 continued on next page*

*Figure 4 continued*

change increase of precursor numbers from fetal to adult stage, as well as fold change of CRU and cell counts. Error bars represent mean ± SD. ns, non-significant, *p < 0.05, **p <0.001.

The online version of this article includes the following source data and figure supplement(s) for figure 4:

**Source data 1.** Original data for plotting *Figure 4*.

**Figure supplement 1.** Analysis related to 5-FU treatment and Confetti fetal induction.

**Figure supplement 1—source data 1.** Original data for plotting *Figure 4—figure supplement 1*.

4 months), suggesting Confetti+-labeled precursors contribute more to post-natal T cells than non-labeled precursors (*Figure 4D*). Therefore, for T cells in E14.5-induced animals, we focus on precursor numbers generated after 3 months of age. The Confetti labeling for E11.5-induced animals remain stable for all timepoints examined (*Figure 4—figure supplement 1C*).

Similar to adult-induction, the distribution of average FP% is stable for fetal-induction, supporting equal proliferation of RFP+ and YFP+ cells (*Figure 4—figure supplement 1D*). Unlike adult-induction, fetal-labeled precursor numbers can be calculated with various combination of FP%, since the expression of CFP introduced at the fetal stages is stable (*Figure 3—figure supplement 1F*). Nonetheless, for consistency, all precursor numbers are calculated with variance of $RFP_{RFP+YFP}$%, the one employed in adult-induction animals.

The resulting precursor number estimates for E11.5 and E14.5-labeled cohorts are similar in all PB cell types except at 2 and 3 months of age, echoing the previous study reporting limited lifelong precursor expansion in the fetal liver (*Figure 4E*; *Ganuza et al., 2022*). Precursor numbers calculated from BM subpopulations are also comparable between the two timepoints (*Figure 4F*). Although E9.5-labeled clones are reported to seed non-uniformly across bones, the precursor numbers calculated from PB myeloid cells do not significantly differ from those calculated from BM MP and HSPC in E14.5-labeled animals (*Figure 4F*; *Bowling et al., 2020*).

For comparison between fetal- and adult-induced animals, we focus on PB myeloid cells, as adult-induced animals have non-saturated labeling in B cells and non-HSPC-rooted Confetti labeling in T cells (*Figure 3B*). Here, we observe a relatively small increase of precursor numbers in adult-induced animals in PB and BM compared to those labeled at the fetal stage (*Figure 4E, F*). The increase between E11.5/E14.5- and adult-labeled precursors is less than 10-fold, similar to a previous report (*Figure 4G*; *Ganuza et al., 2022*).

Together, we confirm minimal to no expansion of lifelong precursors in the fetal liver stage and a minor expansion of expansion of lifelong precursors from fetal liver to adult stage.

## *Fancc*−/− mice have normal numbers of hematopoietic precursors at steady-state

Our approach provides an opportunity to investigate the number of active HSPC precursors in the native environment. To showcase precursor quantification in genetic mouse models, we focus on Fanconi Anemia (FA), the most common inherited BM failure syndrome (*Ceccaldi et al., 2012*). Current FA mouse models exhibit mostly normal adult hematopoiesis at steady state but demonstrate reduced repopulation ability upon BM transplantation (*Carreau et al., 1999*; *Haneline et al., 1999*; *Zhang et al., 2010*; *Du et al., 2015*; *Dubois et al., 2019*; *Zha et al., 2019*). It remains unclear if their steady-state blood production is sustained by a reduced number of precursors, predisposing them to repopulation defects after transplantation.

To quantify precursor number in a mouse model of FA, we generate *Fancc*+/+, *Fancc*+/−, and *Fancc*−/− mice with a single *Confetti* allele and homozygous *HSC-SCL-CreER*T alleles (*HSC-SCL-CreER*T/T*Rosa26*+/ConfettiFancc+/+, hereafter *Fancc*+/+; *HSC-SCL-CreER*T/T*Rosa26*+/ConfettiFancc+/−, hereafter *Fancc*+/−; *HSC-SCL-CreER*T/T*Rosa26*+/ConfettiFancc−/−, hereafter *Fancc*−/−) (*Chen et al., 1996*). Consistent with previous literature, we observe similar PB cell frequencies and blood counts between *Fancc*+/+ and *Fancc*−/− mice (*Figure 5—figure supplement 1A–C*; *Zha et al., 2019*). BM nucleated cell counts, as well as HSPC and MP frequencies, are mostly identical, except for ST-HSC frequencies, which are significantly reduced in *Fancc*−/− mice compared to *Fancc*+/− mice (*Figure 5—figure supplement 1D, E*).

To label hematopoietic precursors in FA animals, Confetti expression was induced at 2 months of age, and FP% was monitored over 7 months (*Figure 5A*). The absence of *Fancc* did not affect Confetti

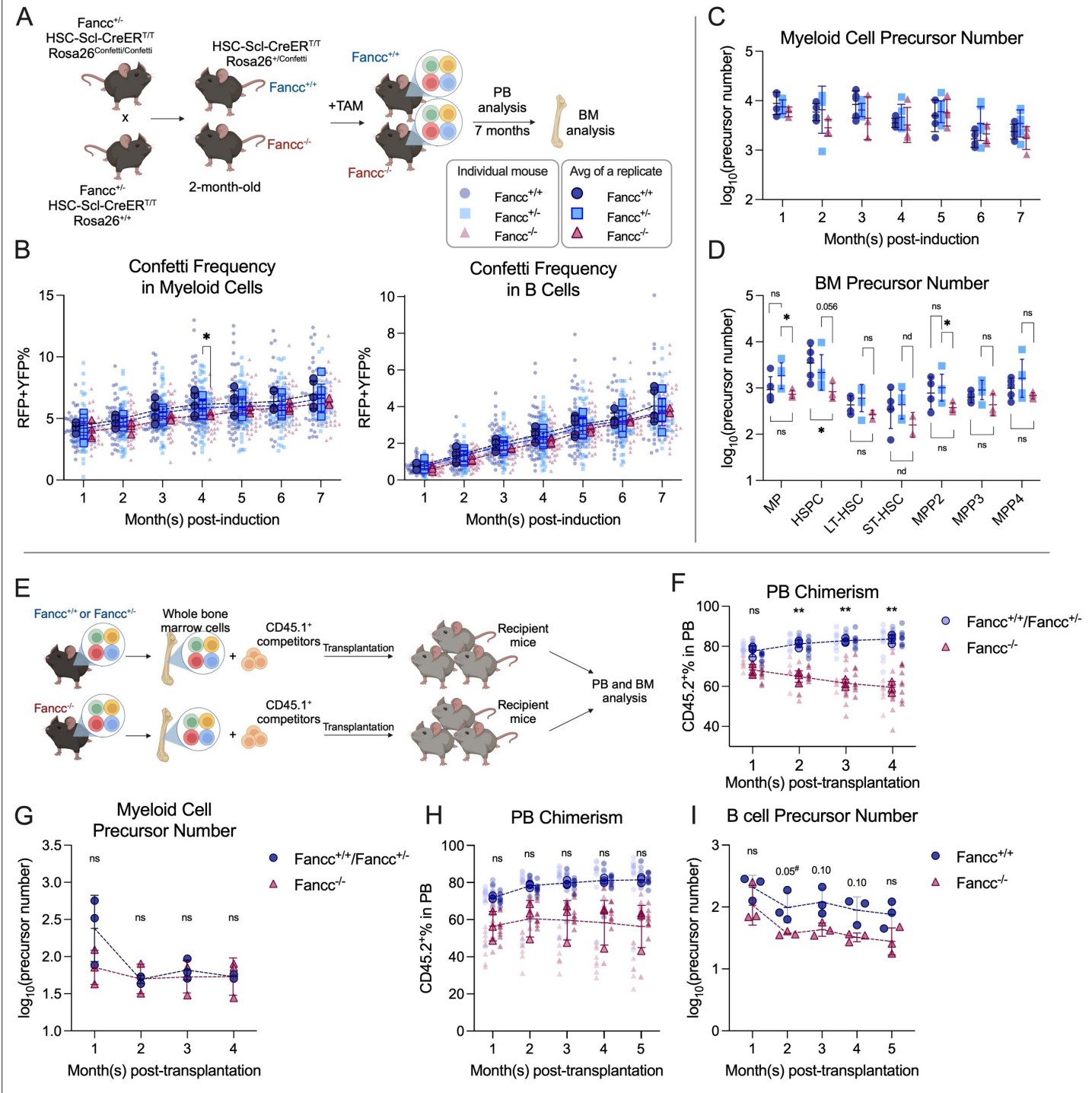

**Figure 5.** Precursor numbers in a mouse model of Fanconi Anemia (FA). (**A**) Experimental workflow to generate *Fancc*⁺/⁺ and *Fancc*⁻/⁻ mice, induce Confetti expression and track precursor number dynamics. Image was created with BioRender.com/f96e500. (**B**) Confetti labeling in peripheral blood (PB) B cells and myeloid cells. Each dot represents one animal. Each outlined dot represents the average value of a replicate containing multiple animals. Dashed lines connect the values for the same replicate. (**C**) The number of PB myeloid precursors. (**D**) The number of bone marrow (BM) precursors. Permutation test was performed. (**E**) Experiment schematic for competitive transplantation. Image was created with BioRender.com/f96e500. (**F**) PB donor chimerism in recipient mice of young FA donors. Each dot represents one animal. Dashed lines connect the average donor chimerism of three replicates. For Fancc⁺/⁺ or Fancc⁺/⁻, *n* = 10 per replicate, *N* = 3 replicates; for Fancc⁻/⁻, *N* = 7–9 per replicate, *N* = 3 replicates. Two-way ANOVA was performed. (**G**) The number of PB myeloid precursors post-transplantation of young FA donors. Dashed lines connect the average precursor numbers. Permutation test was performed. *N* = 3 for each group. (**H**) PB donor chimerism recipient mice of aging FA donor cells. Each dot represents

*Figure 5 continued on next page*

*Figure 5 continued*

one animal. Dashed lines connect the average donor chimerism of three replicates. For both genotype, *N* = 10 per replicate, *N* = 3 replicates. Two-way ANOVA was performed. (**I**) The number of PB B cell precursors post-transplantation of aging FA donors. Dashed lines connect the average precursor numbers. Permutation test was performed. *N* = 3 for each genotype. For (**A–C**), Fancc$^{+/+}$, sample size = 11–17 mice per replicate, *N* = 5 replicates; Fancc$^{+/−}$, sample size = 10–22 mice per replicate, *N* = 7 replicates; Fancc$^{−/−}$, sample size = 9–17 mice per replicate, *N* = 4 replicates. For (**D**), Fancc$^{+/+}$, sample size = 5–9 mice per replicate, *N* = 5 replicates; Fancc$^{+/−}$, sample size = 6–12 mice per replicate, *N* = 5 replicates; Fancc$^{−/−}$, sample size = 6–11 mice per replicate, *N* = 4 replicates. Error bars represent mean ± SD. nd, not determined; ns, non-significant; *$p < 0.05$, **$p < 0.001$. # represents lowest p value possible for permutation test.

The online version of this article includes the following source data and figure supplement(s) for figure 5:

**Source data 1.** Original data for plotting *Figure 5*.

**Figure supplement 1.** Analysis of Fanconi Anemia (FA) mice at steady state.

**Figure supplement 1—source data 1.** Original data for plotting *Figure 5—figure supplement 1*.

**Figure supplement 2.** Analysis of Fanconi Anemia (FA) recipient mice.

**Figure supplement 2—source data 1.** Original data for plotting *Figure 5—figure supplement 2*.

labeling efficiency, Confetti labeling dynamics or average FP% (RFP$_{RFP+YFP}$%), suggesting precursor numbers in *Fancc$^{−/−}$* can be similarly calculated (***Figure 5B***, ***Figure 5—figure supplement 1F***). As in previous adult-induction animals, RFP$_{RFP+YFP}$% was used to estimate precursor numbers. The resulting myeloid and BM precursor numbers were comparable regardless of *Fancc* genotype, albeit *Fancc$^{−/−}$* mice had a slight reduction in HSPC precursor numbers (***Figure 5C, D***). These observations collectively suggest that a normal number of precursors sustains blood production in *Fancc$^{−/−}$* mice.

## The number of *Fancc$^{−/−}$* precursors remains unchanged in mice transplanted with young donor cells

While *Fancc$^{−/−}$* mice have a similar number of precursors as their wildtype counterparts at homeostasis, it is unknown whether the reduced repopulation ability post-transplantation stems from diminished precursor numbers, reduced cell expansion, or a combination of both (***Haneline et al., 1999***). To understand the underlying mechanism, we performed competitive transplantation using BM cells from 3-month-old Confetti-induced *Fancc$^{+/+}$*, *Fancc$^{+/−}$*, or *Fancc$^{−/−}$* mice along with CD45.1$^{+}$ competitor cells (***Figure 5E***, ***Figure 5—figure supplement 2A***). Consistent with previous studies, recipient mice of *Fancc$^{−/−}$* cells show significantly lower donor chimerism in the PB and BM compared to those of *Fancc$^{+/+}$* or *Fancc$^{+/−}$* cells (***Figure 5F***, ***Figure 5—figure supplement 2B***). Despite lower donor chimerism, PB and BM precursor numbers were unaffected in *Fancc$^{−/−}$* recipient mice (***Figure 5G***, ***Figure 5—figure supplement 2C–E***). This suggests that reduced *Fancc$^{−/−}$* cell proliferation instead of fewer active precursors is likely the cause for the reduced repopulation capacity post-transplantation.

## Aging *Fancc$^{−/−}$* mice have reduced lymphoid hematopoietic precursors upon transplantation

Aging *Fancc$^{−/−}$* mice have been reported to develop hematologic neoplasms, resulting in decreased survival (***Cerabona et al., 2016***). To determine if aging *Fancc$^{−/−}$* cells also maintain similar precursor numbers post-transplantation, we competitively transplanted BM cells from 9-month-old Confetti-induced *Fancc$^{+/+}$* or *Fancc$^{−/−}$* mice with CD45.1$^{+}$ competitor cells, a stage when aging *Fancc$^{−/−}$* mice start to show decreased survival (***Cerabona et al., 2016***; ***Figure 5—figure supplement 2F***).

The diminished repopulation ability of *Fancc$^{−/−}$* cells is reaffirmed by lower PB and BM chimerism (***Figure 5H***, ***Figure 5—figure supplement 2I***). While no differences in precursor numbers are noted during the initial post-transplantation period, a slight yet consistent reduction in lymphoid precursors is observed in *Fancc$^{−/−}$* recipient mice at 3–5 months post-transplantation (***Figure 5I***, ***Figure 5—figure supplement 2G, H***). Changes in precursor numbers in the BM of KO recipients are less clear, as we observed high variance in several HSPC subtypes (***Figure 5—figure supplement 2J***). Nonetheless, precursor numbers of MEP, LSK, and ST-HSC showed a consistent reduction. In conclusion, aging *Fancc$^{−/−}$* mice showed a modest but consistent loss of active PB lymphoid precursors post-transplantation, implying decreased lymphoid precursors additionally compromise repopulation capacity as *Fancc$^{−/−}$* mice age.

## Discussion

The polyclonal nature of endogenous hematopoiesis imposes methodological problems on a robust dynamic measurement. Inspired by the XCI studies, we successfully employed the correlation formula informed by binominal distribution to quantify precursor number in native hematopoiesis. Using *HSC-SCL-CreER^T^/Confetti* animals, we estimate thousands of precursors contributing to adult native hematopoiesis, a number comparable to the previous report (*Sun et al., 2014*). This number results from a moderate increase during fetal-to-adult transition and respond dynamically to myeloablation by 5-FU. Applied to a mouse model of inherited BM failure (*Fancc^−/−^* mice), we detected normal precursor numbers at steady state and decreased lymphoid precursors upon transplantation of aging donors.

Although the linear relationship between variance of FP% and precursor numbers has been described, its measurable range has been limited (*Ganuza et al., 2017*). Estimates outside this linear range may erroneously fall within it, making it challenging to distinguish accurate measurements within the range from inaccurate ones beyond it (*Ganuza et al., 2017*). Based on binomial distribution, we expanded the measurable range to encompass the full spectrum of hematopoietic precursors in mice, minimizing the likelihood of inaccuracies. Furthermore, the linear correlation based on binomial distribution can accommodate any labeling system that follows the underlying premises, without the need for the multispectral *Confetti* cassette.

Having established a quantitative measurement, we were initially surprised by the moderate differences between fetal and adult precursors (*Ganuza et al., 2022*). The maximum increase observed (7-fold observed in CMP) was substantially lower than the increase of competitive repopulation units (CRU) from E12 to E16 determined by limited dilution assays (~38-fold) (*Figure 4G*), as well as the escalation from immunophenotype-defined HSPC counts (28- to 47-fold, from 3 to $5 \times 10^3$ at E14.5– $1.4 \times 10^5$ at 2 months old) (*Figure 4G*; *Cosgrove et al., 2021*; *Ema and Nakauchi, 2000*; *Young et al., 2021*; *Mochizuki-Kashio et al., 2020*). It is possible that we overestimated E11.5 precursors, as the numbers of lifelong E11.5 precursors were substantially higher (on average 753 PB myeloid precursor, 512 HSPC precursors) than the one to two repopulation units estimated with transplantation (*Kumaravelu et al., 2002*). Nonetheless, our results were comparable to the ~870 hematopoietic cluster cells observed at E11.5 (*Yokomizo et al., 2022*). We may have underestimated precursor numbers in adult-labeled animals, as most adult HSCs are quiescent (*Wilson et al., 2008*). However, HSC precursor numbers did not increase after induced proliferation by one dose 5-FU treatment (*Figure 4B*). Therefore, we confirm the moderate precursor numbers difference between fetal and adult stages, emphasizing the analysis of hematopoiesis in a native environment. Future studies using different methods to investigate precursor activity locally should validate this result.

Novel methods tracing clones in situ offer new opportunities to study native hematopoiesis, yet most are challenging to apply in genetic models (*Sun et al., 2014*; *Bowling et al., 2020*; *Pei et al., 2017*; *Li et al., 2023*; *Rodriguez-Fraticelli et al., 2018*). Since only two mouse lines are required (*Confetti* and *Cre*, sometimes *Cre* is already included for conditional deletion of alleles), our approach is particularly convenient to study hematopoiesis in mouse models of genetic disorders. Applying precursor measurements to *Fancc^−/−^* mice, we observe a minor decrease in precursor numbers after transplantation of aging donors. Although donor chimerism differences had been linked to differences in precursor numbers, reduced proliferation capacity may also contribute to reduced competitive repopulation capacity (*Harrison et al., 1993*). In this case, definitive precursor number analysis is necessary to differentiate these possibilities.

Currently, concerns regarding clonal restriction in the context of FA gene therapy arises, as the engraftment of gene-corrected stem cells has yielded marginal clinical benefits (*Rio et al., 2019*). For the first time, we demonstrate that FA cells maintain a normal precursor quantity post-transplantation, thereby disproving clonal attrition, including putative homing deficits post-transplantation as a cause in this model (*Zhang et al., 2008*). However, it is imperative to acknowledge that the majority of murine models of FA, including the *Fancc^−/−^* model utilized herein, do not recapitulate the patho-physiology observed in FA patients. Future investigations in other FA mice and studies leveraging FA patient-derived materials will be pivotal in corroborating and validating the findings presented here.

While we implemented a careful data processing procedure, one pitfall of our analyses was that we inferred precursor numbers from their progenies, assuming uniform and linear expansion from precursor to progenies. A recent study showed non-uniform precursor clone sizes, although the level of non-uniformity is low (*Li et al., 2023*). In cases where the non-uniformity is high, according

to mathematical modeling, we measured the major contributors to hematopoiesis (*Stone, 1983*). Another potential caveat is that the relative contribution of Confetti-labeled precursors to blood production compared to non-labeled precursors remains unknown. Future studies using different Cre drivers should validate the precursor numbers for steady-state hematopoiesis.

In summary, we substantially broadened the applicable range of the correlation between variance of FP% and the precursor based on binomial distribution. We discovered thousands of precursors contributing to steady-state adult murine hematopoiesis and validated that fetal-to-adult precursor expansion is indeed limited. This analysis highlights active precursor numbers as an important metric in both normal and genetic mouse models.

## Materials and methods

### Variance modeling with two-color HL-60

HL-60 cells were cultured with IMDM containing 20% fetal bovine serum (Gemini) and 5% penicillin–streptomycin (Gibco) and maintained at $1 \times 10^5$ and $1 \times 10^6$ cells/ml. Mycoplasma tests (Lonza) were performed routinely to rule out mycoplasma contamination. BFP-HL-60 and GFP-HL-60 were generated with lentivirus transduction of pGK-BFP (Genscript) and LeGO-V2 (a gift from Dr. Stefano Rivella lab). For seeding of one to 10,000 cells, sorting of HL-60 mixtures was performed using a BD FACS Aria III. For seeding of 100,000 cells, cell counts and dilution was used. After expansion, HL-60 cells were fixed with BD fixation buffer before Aurora (Cytek) or Cytoflex (Beckman Coulter) analysis.

### Mice

HSC-SCL-CreER$^T$ mice (*Göthert et al., 2005*) were crossed with *Confetti* mice (*Rosa26$^{Brainbow2.1(Confetti)}$*, full name B6.129P2-*Gt(ROSA)26Sor$^{tm1(CAG-Brainbow2.1)Cle}$*/J) to generate *HSC-SCL-CreER$^{T/T}$Rosa26$^{+/Confetti}$* (*HSC-SCL-CreER$^T$/Confetti*) animals. *HSC-SCL-CreER$^T$/Confetti* animals were crossed with *Fancc$^{+/-}$* mice (*Chen et al., 1996*) to generate *HSC-SCL-CreER$^{T/T}$Rosa26$^{+/Confetti}$Fancc$^{+/+}$* (*Fancc$^{+/+}$*), *HSC-SCL-CreER$^{T/T}$Rosa26$^{+/Confetti}$Fancc$^{+/-}$* (*Fancc$^{+/-}$*), and *HSC-SCL-CreER$^{T/T}$Rosa26$^{+/Confetti}$Fancc$^{-/-}$* (*Fancc$^{-/-}$*) mice. *Vav-Cre* was generously offered by Dr. Wei Tong (Children's Hospital of Philadelphia). *Vav-CreRosa26$^{+/Confetti}$* animals were generated by crossing *Vav-Cre* with *Rosa26$^{Confetti}$* mice. All animals are in the B6 (CD45.2) strain background, unless otherwise stated. Six- to twelve-week-old females were used for timed pregnancies. Eight-week-old mice were used for adult Confetti induction. Both female and male mice were used (*Ganuza et al., 2017*). Six- to eight-week-old female B6 CD45.1 mice (B6.SJL-*Ptprc$^a$ Pepc$^b$*/BoyJ, Jackson laboratories) were used as competitors and recipients for transplantation studies. All mice were maintained in the conventional small animal facility at the Children's Hospital of Philadelphia (CHOP). All procedures involving animals were approved by the Institutional Animal Care and Use Committee at the Children's Hospital of Philadelphia.

### Animal identification

Tail snip DNA was extracted using KAPA Express Extract Kit (Roche). Genotyping PCR was performed with HotStarTaq Master Mix (Qiagen) according to the manufacturer's instruction. Genotyping primers used are summarized in *Supplementary file 3*. To determine the zygosity of HSC-SCL-CreER$^T$, qPCR was additionally performed with purified tail snip DNA using SYBR Green Universal Master Mix (Applied Biosystems).

### Animal procedures

For fetal induction, timed matings of HSC-SCL-CreER$^{T/T}$Rosa26$^{Confetti/Confetti}$ mice and HSC-SCL-CreER$^{T/T}$Rosa26$^{+/+}$ mice were set up. The mice were separated the next morning and noon of the day of separation was considered E0.5. Tamoxifen was delivered at 100 mg/kg to the dam orally at E11.5 or E14.5. Pups were C-sectioned and cross-fostered at E18.5 due to reported delivery difficulties caused by tamoxifen treatment (*Lizen et al., 2015*). For mice used for defined ('low' versus 'high') number of transplantation (*Figure 2*), tamoxifen was delivered at 70 mg/kg orally once per day for 14 days at 8 weeks old. For adult-induction (*Figure 3*), tamoxifen was delivered at 70 mg/kg orally once per day for 2 days at 8-week-old. For one dose 5-fluorouracil (5-FU, Sigma) treatment, 5-FU was intraperitoneally injected once 10 days before BM harvest (37-week-old) at 150 mg/kg. To obtain PB for Confetti analysis, mice were anesthetized using isoflurane and retro-orbitally bled or submandibularly bled for

*Fancc$^{-/-}$* mice with occasional congenital eye defects (*Joly et al., 2022*) for 1 capillary of blood (50 µl). For PB counts, blood was collected in EDTA tubes using similar bleeding methods and was analyzed by the Translational Core Lab (Children's Hospital of Philadelphia).

## BM sample processing

To collect BM cells, mice were euthanized by $CO_2$ inhalation. Tibia, femur and pelvis were dissected, and the BM cells were flushed with 26-gauge needles. The single-cell suspension generated with 18-gauge needles was then filtered through 70 µm strainers. Erythrocytes in the BM cells were hemolyzed by red blood cell lysis buffer before antibody staining or transplantation. For stem cell enrichment, BM cells were further lineage depleted using EasySep Mouse Hematopoietic Progenitor Cell Isolation Kit (STEMCELL Technologies).

## Flow cytometry analysis

PB and BM cells were analyzed using Aurora (Cytek) or BD FACS Aria III and the flow cytometry data were analyzed using FlowJo (Tree Star). The combinations of the following cell surface markers were used to define the PB populations: myeloid cells: $CD11b^+$ or Gr-$1^+$; T-cell: $CD3\varepsilon^+$; B-cell: $B220^+$. The following combinations of cell surface markers were used to define the BM stem and progenitor cells (Lineage/Lin: CD11b, Gr-1, B220, CD3ε, Ter119): LTHSC: Lin$^-$c-Kit$^+$Sca1$^+$Flk2$^-$CD150$^+$CD48$^-$; MPP2: Lin$^-$c-Kit$^+$Sca1$^+$Flk2$^-$CD150$^+$CD48$^+$; MPP3: Lin$^-$c-Kit$^+$Sca1$^+$Flk2$^-$CD150$^-$CD48$^+$; MPP4: Lin$^-$c-Kit$^+$Sca1$^+$Flk2$^+$CD150$^-$CD48$^+$; STHSC: Lin$^-$c-Kit$^+$Sca1$^+$Flk2$^-$CD150$^-$CD48$^-$; MEP: Lin$^-$c-Kit$^+$Sca1$^-$CD34$^-$CD16/32$^-$; CMP: Lin$^-$c-Kit$^+$Sca1$^-$CD34$^{mid}$CD16/32$^{mid}$; GMP: Lin$^-$c-Kit$^+$Sca1$^-$CD34$^+$CD16/32$^+$. For BM stem and progenitor cell analysis, DAPI (Biolegend) was used to distinguish dead cells. Representative examples of flow cytometry gating can be found in *Figure 2—figure supplement 1C*. The antibodies were used at optimized dilutions listed in *Supplementary file 1*.

## BM transplantation

The day before transplantation, female CD45.1 recipient mice were lethally irradiated (5.2 Gy × 2, 3 hr apart) using an X-ray irradiator (Precision). On the day of transplantation, BM cells from donor mice (CD45.2$^+$) were collected under sterile conditions as described, RBC-lysed and counted for cell number. For *Figure 2* (transplantation of defined precursor numbers), donor mice were induced to express Confetti as described; donor BM cells were (non-competitively) injected into irradiated recipient mice; each matched high- and low- precursor number group received donor BM cells pooled from three to five mice. For transplantation of young *Fancc* mice, donor mice were induced to express Confetti at E14.5; $1.5 \times 10^6$ donor BM cells were mixed with $2.5 \times 10^5$ CD45.1 supporting BM cells and injected into irradiated recipient mice via tail vein. For transplantation of aging Fancc mice, donor mice were induced to express Confetti at 2 months of age; $2 \times 10^6$ donor BM cells were mixed with $5 \times 10^5$ CD45.1 supporting BM cells.

## Derivation of the correlation between variance of Confetti FP% and precursor numbers based on binomial distribution

When a random variable adheres to binomial distribution, studies have established that the following equation holds true:

$$n = \frac{p\,(1-p)}{\sigma^2}$$

where *n* signifies the number of precursors, *p* represents the probability of an individual being one of the FPs (e.g., RFP), and $\sigma^2$ denotes the variance of specific FP% (e.g., $\sigma_{RFP\%}{}^2$) (*Wallis et al., 1975*). In experiments, *p* will be estimated with the average FP% in the Confetti$^+$ cells (e.g., RFP$_{mean}$%), and $\sigma^2$ will be estimated with the variance of FP% among a group of individual or mice (e.g., $\widehat{\sigma_{RFP}}^2$). Consequently, the estimation of precursors number *n* is calculated using the following equation:

$$\hat{n} = \frac{\widehat{FP_{mean}}\% \times \left(1 - \widehat{FP_{mean}}\%\right)}{\widehat{\sigma_{FP}}^2}$$

A logarithmic transformation can then be performed, allowing us to establish a linear relationship between the variance of FP% and the number of precursors:

$$\lg\left(\hat{n}\right) = \lg\left[\widehat{FP_{mean}}\% \times \left(1 - \widehat{FP_{mean}}\%\right)\right] - 2 \times \lg\left(\widehat{\sigma_{FP}}\right)$$

In a previous study (*Ganuza et al., 2017*),

$$\sigma_{FP} = CV_{FP} \times FP_{mean}\%$$

Therefore,

$$\lg\left(\hat{n}\right) = \lg\left[\frac{\left(1 - \widehat{FP_{mean}}\%\right)}{\widehat{FP_{mean}}\%}\right] - 2 \times \lg\left(\widehat{CV_{FP}}\right)$$

### Resampling to determine sample size per replicate

FP% data generated from HL-60 was used for re-sampling. For each seeded number, FP% was resampled for different sample sizes from all the FP% with replacement. Variance of estimated $n$ was then calculated by the standard deviation of the estimated precursor numbers generated from the same sample size. Relative error was determined by dividing variance of estimated $n$ with the average of the estimated $n$. Refer to 'varying well numbers.rmd' for detailed R code.

### Simulation to determine the effect of varying FP$_{mean}$% on correlation between variance of FP% and precursor numbers

Simulation of binomial distribution of varying FP$_{mean}$% (probability of being a FP) was performed in R, generating corresponding FP% values used for variance calculation. The correlation between variance and precursor number $n$ was then determined by linear correlation, and the slopes and intercepts were compared to each other. Refer to 'Simulation of varying FPmean_percent.rmd' for detailed R code.

### Simulation to determine the effect of precursor numbers following Poisson post-transplantation

Precursor numbers in individual samples were simulated to follow a Poisson distribution, where the mean of precursor numbers is the expected precursor numbers (expected $n$). The induction of an FP in each precursor was then simulated by random assignment, where probability of being an FP was set to be FP$_{mean}$%. For each expected precursor numbers, variance of FP% among samples was then calculated, and *equation 2* was used to estimate precursor numbers (estimated $n$). The correlation between expected $n$ and estimated $n$ was then plotted. Refer to 'Double layer binomial simulation. rmd' for detailed R code.

### Data processing and normalization

PB cell subset frequencies were normalized to the total % of myeloid cells, T cell, and B cell, to avoid an underestimate due to incomplete RBC lysis. For transplant animals, PB CD45.1[+]% and CD45.2[+]% are normalized to total CD45.1[+]% and CD45.2[+]% to avoid an underestimate due to incomplete RBC lysis. After normalization of cell frequencies, the sum of Confetti% and FP$_{Conf}$% (diving FP$_{PB}$% or FP$_{BM}$% by sum of Confetti%) were then calculated for each cohort. At each step, potential outliers were removed based on Tukey method. The variance of FP$_{Conf}$% and average FP$_{Conf}$% are then fitted into *equation 2* to calculate precursor numbers if sample size is at least five. The precursor number is then used to compared with the minimum flow cytometry recorded event of samples. If the estimated number is higher than the minimum flow count of samples, the sample with the minimum flow count will be excluded, and the calculation is performed again on the rest of samples. After exclusion, the resulting precursor number will be compared again with minimum flow count of samples, until it is lower than the minimum flow count or sample size is smaller than five.

## Statistical analysis

### Statistical significance for precursor number estimates

As the distribution of precursor numbers is not predetermined, to compare mean precursor number differences between two conditions from a limited number of biological replicates, we employed permutation test. For unpaired permutation test, if there are three to five biological replicates per condition, the smallest p values possible are 0.05–0.004 ($1/\binom{6}{3}$–$1/\binom{5}{10}$). For paired permutation test, if there are three to five biological replicates per condition, the smallest p value possible are 0.125–0.03125 ($1/2^3$–$1/2^5$). Thus, for three to five biological replicates per condition, even though some of the p values may not reach the commonly used alpha level (0.05), it still represents substantial number differences. For those p values that were lowest possible but did not reach the alpha level, we specifically labeled with a '#' in the figures and legends.

### Statistical analyses of other data

All other two-sample statistical analyses were performed using Student's $t$ test, if the sample was normally distributed, or Welch's $t$ test, when the sample was not normally distributed ($F$ test). For multiple comparison, one- or two-way ANOVA was used.

# Acknowledgements

Work was supported by R01-HL150882. We thank Florin Tuluc (CHOP Flow Cytometry Core) and Jessica Gucwa (Cytek bioscience) for assistance in flow cytometry; Kaosheng Lv (CHOP) for preparation of reagents; Zilu Zhou (Penn) for help in statistical analysis; Nancy Speck (Penn), Wei Tong (Penn), Julia Warren (Penn), Ding-wen (Roger) Chen (CHOP), Stephanie N Hurwitz (Penn), Hua Qing (Genentech), and Hui Chen (Penn) for in-depth discussion.

# Additional information

## Funding

| Funder | Grant reference number | Author |
| --- | --- | --- |
| National Heart, Lung, and Blood Institute | HL150882 | Peter Kurre |

The funders had no role in study design, data collection, and interpretation, or the decision to submit the work for publication.

## Author contributions

Suying Liu, Conceptualization, Data curation, Software, Formal analysis, Investigation, Methodology, Writing - original draft, Writing - review and editing; Sarah E Adams, Investigation, Methodology, Project administration; Haotian Zheng, Validation, Methodology; Juliana Ehnot, Seul K Jung, Greer Jeffrey, Theresa Menna, Investigation, Methodology; Louise Purton, Conceptualization, Writing - review and editing; Hongzhe Lee, Supervision, Investigation, Methodology; Peter Kurre, Conceptualization, Supervision, Funding acquisition, Investigation, Methodology, Project administration, Writing - review and editing

## Author ORCIDs

Suying Liu ⓘ https://orcid.org/0000-0003-3947-8151
Peter Kurre ⓘ https://orcid.org/0000-0003-2747-0099

Reviewer #1 (Public review): https://doi.org/10.7554/eLife.97504.3.sa1
Reviewer #2 (Public review): https://doi.org/10.7554/eLife.97504.3.sa2
Reviewer #3 (Public review): https://doi.org/10.7554/eLife.97504.3.sa3
Author response https://doi.org/10.7554/eLife.97504.3.sa4

## Additional files

### Supplementary files
- Supplementary file 1. Antibody dilutions used for flow cytometry.
- Supplementary file 2. The number of animals used for each figure/experiment.
- Supplementary file 3. Genotyping primer sequences.
- MDAR checklist

### Data availability
The original Confetti% measured in the PB and the BM and R code to analyze and reproduce all the results, numeric and figures can be found at https://doi.org/10.5281/zenodo.8222789. The re-analyzed data from a previous study can be found in the online version of papers (*Ganuza et al., 2017*).

The following dataset was generated:

| Author(s) | Year | Dataset title | Dataset URL | Database and Identifier |
|---|---|---|---|---|
| Suying L, Sarah A, Haotian E, Jung J, Greer J, Theresa M, Louise EP, Hongzhe L, Peter K | 2023 | VariClone Resolves Population-level Clonal Structure in Native and Stress Hematopoiesis | https://doi.org/10.5281/zenodo.8222789 | Zenodo, 10.5281/zenodo.8222789 |

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

# Appendix 1

## Appendix 1—key resources table

| Reagent type (species) or resource | Designation | Source or reference | Identifiers | Additional information |
|---|---|---|---|---|
| Strain, strain background (*Mus musculus*, B6) | CD45.1 | Jackson laboratories | B6.SJL-*Ptprc^a Pepc^b*/BoyJ | |
| Strain, strain background (*M. musculus*, B6) | CD45.2 | Jackson laboratories | C57BL/6J | |
| Genetic reagent (*M. musculus*, B6) | *HSC-SCL-CreER^T/Confetti* | Dr. Louise Purton | C57BL/6-Tg(Tal1-cre/ERT)42-056Jrg/J and B6.129P2-*Gt(ROSA)26Sor^{tm1(CAG-Brainbow2.1)Cle}*/J | |
| Genetic reagent (*M. musculus*, B6) | *Fancc^{+/−}* | **Cerabona et al., 2016, Chen et al., 1996** | | |
| Genetic reagent (*M. musculus*, B6) | Vav-Cre | Dr. Tong Wei | B6.Cg-Tg(VAV1-cre)1Graf/MdfJ | |
| Cell line (*Homo sapiens*) | HL-60 | ATCC | | |
| Cell line (*H. sapiens*) | 293T | ATCC | | To generate lentivirus |
| Antibody | FcBlock (Rat monoclonal, clone 93) | Biolegend | Cat:101320; RRID:AB_1574975 | (1:500) |
| Antibody | APC anti-B220 (Rat monoclonal, clone RA3-6B2) | Biolegend | Cat:103212; RRID:AB_312997 | Dilution depends on experiment, see **Supplementary file 1** |
| Antibody | APC anti-CD3ε (Armenian Hamster monoclonal, clone 145-2C11) | Biolegend | Cat:100311; RRID:AB_389300 | Dilution depends on experiment, see **Supplementary file 1** |
| Antibody | APC anti-CD11b (Rat monoclonal, clone M1/70) | Biolegend | Cat:101212; RRID:AB_312795 | Dilution depends on experiment, see **Supplementary file 1** |
| Antibody | APC anti-Gr-1 (Rat monoclonal, clone RB6-8C5) | Biolegend | Cat:108412; RRID:AB_313377 | Dilution depends on experiment, see **Supplementary file 1** |
| Antibody | APC anti-TER119 (Rat monoclonal, clone TER119) | Biolegend | Cat:116212; RRID:AB_313713 | Dilution depends on experiment, see **Supplementary file 1** |
| Antibody | APC anti-Ly6C (Rat monoclonal, clone HK 1.4) | Biolegend | Cat:128015; RRID:AB_1732076 | (1:320) |
| Antibody | APC anti-CD4 (Rat monoclonal, clone GK1.5) | Biolegend | Cat:100411; RRID:AB_312696 | (1:100) |
| Antibody | Alexa Fluor 700 anti-CD48 (Armenian Hamster monoclonal, clone HM48-1) | Biolegend | Cat:103426; RRID:AB_10612755 | (1:400) |
| Antibody | Alexa Fluor 700 anti-CD34 (Rat monoclonal, clone RAM34) | Invitrogen | Cat:56-0341-82; RRID:AB_493998 | (1:40) |
| Antibody | Alexa Fluor 700 anti-NK1.1 (Mouse monoclonal, clone PK136) | Biolegend | Cat:108729; RRID:AB_2074426 | (1:100) |
| Antibody | APC/Cyanine7 anti-Sca-1 (Rat monoclonal, clone D7) | Biolegend | Cat:108126; RRID:AB_10645327 | Dilution depends on experiment, see **Supplementary file 1** |
| Antibody | APC/Cyanine7 anti-CD45.2 (Mouse monoclonal, clone 104) | Biolegend | Cat:109824; RRID:AB_830789 | Dilution depends on experiment, see **Supplementary file 1** |
| Antibody | APC/Cyanine7 anti-CD8 (Rat monoclonal, clone 53-6.7) | Biolegend | Cat:100711; RRID:AB_312750 | (1:200) |
| Antibody | PE/Cyanine7 anti-Sca-1 (Rat monoclonal, clone D7) | Biolegend | Cat:108114; RRID:AB_493596 | (1:800) |
| Antibody | PE/Cyanine7 anti-Ly6G (Rat monoclonal, clone 1A8) | Biolegend | Cat:127617; RRID:AB_1877261 | (1:800) |
| Antibody | Pacific Blue anti-B220 (Rat monoclonal, clone RA3-6B2) | Biolegend | Cat:103227; RRID:AB_492876 | (1:200) |
| Antibody | Pacific Blue anti-CD3ε (Armenian Hamster monoclonal, clone 145-2C11) | Biolegend | Cat:100334; RRID:AB_2028475 | (1:200) |

*Appendix 1 Continued on next page*

*Appendix 1 Continued*

| Reagent type (species) or resource | Designation | Source or reference | Identifiers | Additional information |
|---|---|---|---|---|
| Antibody | PerCP anti-CD11b (Rat monoclonal, clone M1/70) | Biolegend | Cat:101229; RRID:AB_2129375 | (1:100) |
| Antibody | Brilliant Violet 421 anti-CD135 (Rat monoclonal, clone A2F10) | Biolegend | Cat:135314; RRID:AB_2562339 | (1:40) |
| Antibody | Brilliant Violet 421 anti-CD45.1 (Mouse monoclonal, clone A20) | Biolegend | Cat:110732; RRID:AB_2562563 | (1:400) |
| Antibody | Brilliant Violet 421 anti-CD41 (Rat monoclonal, clone MWRReg30) | Biolegend | Cat:133911; RRID:AB_10960744 | (1:100) |
| Antibody | Brilliant Violet 510 anti-IA-IE (Rat monoclonal, clone M5/114.152) | Biolegend | Cat:107635; RRID:AB_2561397 | (1:100) |
| Antibody | Brilliant Violet 605 anti-CD150 (Rat monoclonal, clone TC15-12F12.2) | Biolegend | Cat:115927; RRID:AB_11204248 | (1:60) |
| Antibody | Brilliant Violet 650 anti-CD45.2 (Mouse monoclonal, clone 104) | Biolegend | Cat:109836; RRID:AB_2563065 | Dilution depends on experiment, see *Supplementary file 1* |
| Antibody | Brilliant Violet 711 anti-CD45.1 (Mouse monoclonal, clone A20) | Biolegend | Cat:110739; RRID:AB_2562605 | Dilution depends on experiment, see *Supplementary file 1* |
| Antibody | Brilliant Violet 711 anti-CD150 (Rat monoclonal, clone TC15-12F12.2) | Biolegend | Cat:115941; RRID:AB_2629660 | (1:100) |
| Antibody | Brilliant Violet 711 anti-CD16/32 (Rat monoclonal, clone 93) | Biolegend | Cat:101337; RRID:AB_2565637 | (1:320) |
| Antibody | Brilliant Violet 711 anti-B220 (Rat monoclonal, clone RA3-6B2) | Biolegend | Cat:103255; RRID:AB_2563491 | (1:160) |
| Antibody | Brilliant Violet 785 anti-CD127 (Rat monoclonal, clone A7R34) | Biolegend | Cat:135037; RRID:AB_2565269 | (1:80) |
| Antibody | Brilliant Violet 785 anti-CD11c (Armenian Hamster monoclonal, clone N418) | Biolegend | Cat:117335; RRID:AB_11219204 | (1:50) |
| Antibody | BUV395 Anti-Mouse CD117 (Rat monoclonal, clone 2B8) | BD Horizon | Cat:564011; RRID:AB_2738541 | Dilution depends on experiment, see *Supplementary file 1* |
| Recombinant DNA reagent | pGK-BFP | GenScript | | For producing lentivirus to infect HL-60 |
| Recombinant DNA reagent | LeGO-V2 | Dr. Stefano Rivella lab | | For producing lentivirus to infect HL-60 |
| Sequence-based reagent | mScl/Tal1 F | Dr. Louise Purton | HSC-SCL-Cre-Genotyping primers | CAACAACAACCGGGTGAAGA |
| Sequence-based reagent | 1260_1 (TACONIC ctrl) | Dr. Louise Purton | HSC-SCL-Cre-Genotyping primers | GAGACTCTGGCTACTCATCC |
| Sequence-based reagent | Mx-Cre JH43 (Cre spec.) | Dr. Louise Purton | HSC-SCL-Cre-Genotyping primers | CTTGCACCATGCCGCCCACGAC |
| Sequence-based reagent | 1260_2 (TACONIC ctrl) | Dr. Louise Purton | HSC-SCL-Cre-Genotyping primers | CCTTCAGCAAGAGCTGGGGAC |
| Sequence-based reagent | 1341 (Tg F) | Jackson Laboratory | *Confetti* mouse genotyping primers | GAA TTA ATT CCG GTA TAA CTT CG |
| Sequence-based reagent | oIMR8545 (WT F) | Jackson Laboratory | *Confetti* mouse genotyping primers | AAA GTC GCT CTG AGT TGT TAT |
| Sequence-based reagent | oIMR8916 (common) | Jackson Laboratory | *Confetti* mouse genotyping primers | CCA GAT GAC TAC CTA TCC TC |
| Sequence-based reagent | WT | Ref 43, *Chen et al., 1996* | Fancc-Genotyping primers | GAG GAA ACG CCA CAT TTC AG |
| Sequence-based reagent | mutant | Ref 43, *Chen et al., 1996* | Fancc-Genotyping primers | ACG AGA TCA GCA GCC TCT GT |
| Sequence-based reagent | Common Reverse | Ref 43, *Chen et al., 1996* | Fancc-Genotyping primers | AGG TCT GGA GAA ATG GCT CA |
| Commercial assay or kit | EasySep Mouse Hematopoietic Progenitor Cell Isolation Kit | StemCell Technologies | Cat:19856 | |

*Appendix 1 Continued on next page*

*Appendix 1 Continued*

| Reagent type (species) or resource | Designation | Source or reference | Identifiers | Additional information |
|---|---|---|---|---|
| Chemical compound, drug | 5-FU (Fluorouracil) | Sigma | SKU: PHR1227-500MG | |
| Chemical compound, drug | DAPI (4',6-Diamidino-2-Phenylindole, Dilactate) | Biolegend | Cat:422801 | |
| Software, algorithm | Prism 9 | GraphPad | https://www.graphpad.com/scientific-software/prism/ | |
| Software, algorithm | FlowJo v10 | FlowJo | https://www.flowjo.com/solutions/flowjo | |
| Software, algorithm | R studio | Posit Software | https://posit.co/download/rstudio-desktop/ | |
| Software, algorithm | R code to analyze all the data | This paper | https://doi.org/10.5281/zenodo.8222789 | |

