## [Editor Report · eLife Assessment]

This **important** study by Liu and colleagues uses lineage tracing of hematopoietic stem and progenitor cells in situ to infer the clonal dynamics of adult hematopoiesis. The authors apply a new mathematical analysis framework enabling a wider range of clonal estimation and the revised study (1) provides evidence of polyclonal adult hematopoiesis, (2) provides insights on clonal dynamics during fetal liver hematopoiesis, and (3) reveals unexpectedly high polyclonality in a mouse model of bone marrow failure (Fanconi anemia), arguing against the prevalent views of clonal attrition in this context. The evidence in this extensively revised and improved study is **compelling**, with methods, data and analyses more rigorous than the current state-of-the-art, which will be of broad interest not only to stem cell and developmental biologists working on hematopoiesis, but also to researchers working on other systems.

---

## [Referee Report · Reviewer #1 (Public review)]

Previous studies have used a randomly induced label to estimate the number of hematopoietic precursors that contribute to hematopoiesis. In particular, the McKinney-Freeman lab established a measurable range of precursors of 50-2500 cells using random induction of one of the 4 fluorescent proteins (FPs) of a Confetti reporter in the fetal liver to show that hundreds of precursors establish lifelong hematopoiesis. In the presented work, Liu and colleagues aim to extend the measurable range of precursor numbers previously established and enable measurement in a variety of contexts beyond embryonic development. To this end, the authors investigated whether the random induction of a given Confetti FP follows the principles of binomial distribution such that the variance inversely correlates with the precursor number. The authors validated their hypothesis and identified sampling conditions to minimize experimental error using a simplified in vitro system. They use tamoxifen-inducible Scl-CreER, active in hematopoietic stem and progenitor cells (HSPCs), to induce Confetti labeling and investigate whether they could extend their model to cell numbers below 50 with in vivo transplantation of high versus low numbers of Confetti total bone marrow (BM) cells. The data generated are generally robust. While the lower and upper limits of the model may show some small error or have not yet been completely validated experimentally, it extends the measurable range of precursor from 15 - 10^5 cells. The authors then apply their model to estimate the number of hematopoietic precursors that contribute to hematopoiesis in a variety of contexts including adult steady state, fetal liver, following myeloablation, and a genetic model of Fanconi anemia.

Their data highlight the importance of estimating precursor numbers and not just donor frequency in transplantation settings and show that native hematopoiesis is highly polyclonal. Their data also confirm previous findings from Ganuza et al, 2022 that demonstrate no major expansion of precursors between E11.5 - E14.5. Finally, their work reveals intact Fancc-/-precursor numbers following transplantation, suggesting that the observed reduced chimerism is due to defects in cell proliferation.

The conclusions are generally sound and based on high-quality data. As the authors note, future studies should validate the model using alternative Cre-drivers to exclude any potential functional difference between labelled and non-labelled cells. Although this system does not permit tracing of individual clones, the modeling presented allows measurements of clonal activity covering nearly the entire HSPC population (as recently estimated by Cosgrove et al, 2021) and can be applied to a wide range of in vivo contexts with relative ease.

---

## [Referee Report · Reviewer #2 (Public review)]

The manuscript is well written, with beautiful and clear figures, and both methods and mathematical models are clear and easy to understand. Since 2017, Mikel Ganuza, Shannon McKinney-Freeman et al have been using these Confetti approaches that rely on calculating the variance across independent biological replicates as a way to infer clonal dynamics. This is a powerful tool and it is a pleasure to see it being implemented in more labs around the world. One of the cool novelties of the current manuscript is using a mathematical model (based on a binomial distribution) to avoid directly regressing the Confetti labeling variance with the number of clones (which only has linearity for a small range of clone numbers). As a result, this current manuscript of Liu et al. methodologically extends the usability of the Confetti approach, allowing them more precise and robust quantification.

They then use this model to revisit some questions from various Ganuza et al. papers, validating most of their conclusions. The application to the clonal dynamics of hematopoiesis in a model of Fanconi anemia (Fancc mice) is very much another novel aspect, and shows the surprising result that clonal dynamics are remarkably similar to the wild-type (in spite of the defect that these Fancc HSCs have during engraftment).

Overall, the manuscript succeeds at what it proposes to do, stretching out the possibilities of this Confetti model, which I believe will be useful for the entire community of stem cell biologists, and possibly make these assays available to other stem cell regenerating systems.

The revised version has incorporated the reviewer suggestions, strengthening the solidity of the arguments and statements, and highlighting alternative interpretations. My comments were addressed in full.

---

## [Referee Report · Reviewer #3 (Public review)]

The paper presents a solid method for quantifying hematopoietic precursors using statistical variance as a proxy, providing valuable insights into hematopoietic dynamics across different physiological and pathological scenarios. The findings are pivotal for understanding hematopoietic dynamics. The strength of the evidence is convincing and acknowledges limitations such as the binomial assumption and the need of tools to measure clonality.

Liu et al. focus on a mathematical method to quantify active hematopoietic precursors in mice using Confetti reporter mice combined with Cre-lox technology. The paper explores the hematopoietic dynamics in various scenarios, including homeostasis, myeloablation with 5-fluorouracil, Fanconi anemia (FA), and post-transplant environments. The key findings and strengths of the paper include (1) precursor quantification: The study develops a method based on the binomial distribution of fluorescent protein expression to estimate precursor numbers. This method is validated across a wide dynamic range, proving more reliable than previous approaches that suffered from limited range and high variance outside this range; (2) dynamic response analysis: The paper examines how hematopoietic precursors respond to myeloablation and transplantation; (3) application in disease models: The method is applied to the FA mouse model, revealing that these mice maintain normal precursor numbers under steady-state conditions and post-transplantation, which challenges some assumptions about FA pathology. Despite the normal precursor count, a diminished repopulation capability suggests other factors at play, possibly related to cell proliferation or other cellular dysfunctions. In addition, the FA mouse model showed a reduction in active lymphoid precursors post-transplantation, contributing to decreased repopulation capacity as the mice aged. The authors are aware of the limitation of the assumption of uniform expansion. The paper assumes a uniform expansion from active precursor to progenies for quantifying precursor numbers. This assumption may not hold in all biological scenarios, especially in disease states where hematopoietic dynamics can be significantly altered. If non-uniformity is high, this could affect the accuracy of the quantification. Overall, the study underscores the importance of precise quantification of hematopoietic precursors in understanding both normal and pathological states in hematopoiesis, presenting a robust tool that could significantly enhance research in hematopoietic disorders and therapy development. This manuscript would be interesting to the readers of eLife.

---

## [Author Response]

The following is the authors’ response to the original reviews.

**Public Reviews:**

**Reviewer #1 (Public Review):**
Previous studies have used a randomly induced label to estimate the number of hematopoietic precursors that contribute to hematopoiesis. In particular, the McKinneyFreeman lab established a measurable range of precursors of 50-2500 cells using random induction of one of the 4 fluorescent proteins (FPs) of a Confetti reporter in the fetal liver to show that hundreds of precursors establish lifelong hematopoiesis. In the presented work, Liu and colleagues aim to extend the measurable range of precursor numbers previously established and enable measurement in a variety of contexts beyond embryonic development. To this end, the authors investigated whether the random induction of a given Confetti FP follows the principles of binomial distribution such that the variance inversely correlates with the precursor number. They tested their hypothesis using a simplified 2-color in vitro system, paying particular attention to minimizing sources of experimental error (elimination of outliers, sample size, events recorded, etc.) that may obscure the measurement of variance. As a result, the data generated are robust and show that the measurable range of precursors can be extended up to 105 cells. They use tamoxifen-inducible Scl-CreER, which is active in hematopoietic stem and progenitor cells (HSPCs) to induce Confetti labeling, and investigated whether they could extend their model to cell numbers below 50 with in vivo transplantation of high versus low numbers of Confetti total bone marrow (BM) cells. The premise of binomial distribution requires that the number of precursors remains constant within a group of mice. The rare frequency of HSPCs in the BM means that the experimentally generated "low" number recipient animals showed some small variability of seeding number, which does not follow the requirement for binomial distribution. While variance due to differences in precursor numbers still dominates, it is unclear how accurate estimated numbers are when precursor numbers are low (<10).

According to our simulation, the differences between estimated numbers and the corresponding expected numbers are more profound at numbers below 10, but they are still relatively small. Since Figure S4A is in log-scale, it might be difficult for readers to appreciate the magnitude in difference from the graph. We plan to add a linear scale figure to Figure S4A for better visualization of the absolute value differences (left). We also plan to provide an additional graph quantifying the value differences between estimated and expected values for numbers below 15 (right). From both graphs, the maximum difference between estimated n and expected n occurs at 10 precursor numbers (estimated as 7.6). We admit that these numbers are not numerically the same, and some minor correction of the formula may be needed if a very accurate absolute number is warrant. However, we also want to emphasize that 1. most estimated n values are within 25% range of the expected n; 2. despite the minor discrepancy, the estimated n is still highly correlated with the expected n, so the comparison between different precursor numbers was not affected.

**Author response image 1. sa4fig1:** 

The authors then apply their model to estimate the number of hematopoietic precursors that contribute to hematopoiesis in a variety of contexts including adult steady state, fetal liver, following myeloablation, and a genetic model of Fanconi anemia. Their modeling shows:- thousands of precursors (~2400-2600) contribute to adult myelopoiesis, which is in line with results from a previous study (Sun et al, 2014).- myeloablation (single dose 5-FU), while reducing precursor numbers of myeloid progenitors and HSPCs, was not associated with a reduction in precursor numbers of LTHSCs.- no major expansion of precursor number in the fetal liver derived from labeling at E11.5 versus E14.5, consistent with recent findings from Ganuza et al, 2022.- normal precursor numbers in Fancc-/- mice at steady state and from competitive transplantation of young Fancc-/- BM cells, suggesting that reduced Fancc-/- cell proliferation may underlie the reduced chimerism upon transplantation.- reduced number of lymphoid precursors following transplantation of BM cells from 9month-old Fancc-/- animals (beyond this age animals have decreased survival).Although this system does not permit the tracing of individual clones, the modeling presented allows measurements of clonal activity covering nearly the entire HSPC population (as recently estimated by Cosgrove et al, 2021) and can be applied to a wide range of in vivo contexts with relative ease. The conclusions are generally sound and based on high-quality data. Nevertheless, some results could benefit from further explanation or discussion:- The estimated number of LT-HSCs that contribute to myelopoiesis is not specifically provided, but from the text, it would be calculated to be 1958/5 = ~391. Data from Busch et al, 2015 suggest that the number of differentiation-active HSCs is 5.2x103, which is considered the maximum limit. There is nevertheless a more than 10-fold difference between these two estimates, and it is unclear how this discrepancy arises.

First, we would like to clarify a sentence in the manuscript.

“The average myeloid precursor number at the time of BM analysis (1958) matched the average precursor number calculated from BM myeloid progenitors (MP, Lin-Sca-1-cKit+) and HSPCs (1773 and 1917), but it was five-fold higher than that of LT-HSC (Figure 3E).”

In this sentence, we compared the number of precursors calculated from peripheral blood myeloid cells to the those calculated from BM myeloid progenitor, HSPC and LT-HSC. However, we did not intend to imply that those precursors numbers calculated from HSPC and LT-HSC specifically contribute to myelopoiesis. To avoid misunderstanding, we propose to change this sentence to read:

“The average precursor number calculated from PB myeloid cells at the time of BM analysis (1958) matched those calculated from BM myeloid progenitors (MP, Lin-Sca-1-cKit+) and HSPCs (1773 and 1917), but it was fivefold higher than that of LT-HSC (Figure 3E).”

Nonetheless, we appreciate the reviewers’ comment on the gap between the precursor numbers of LT-HSC and the number of differentiation-active HSCs reported in Busch et al, 2015. We propose the following explanation:

First of all, precursor numbers reflect LT-HSC self-renewal by symmetric division and maintenance by asymmetric division but not differentiation. To compare the number of differentiation-active LT-HSC, precursor numbers measured from differentiated progeny (progenitors) is a better choice. As our system does not differentiate the origin of a precursor, measuring the precursor number of differentiation-active LT-HSC is difficult, since progenitors may also derive from other long-lived MPPs. However, if we assume that most divisions of LT-HSC are asymmetric division, generating one LT-HSC and one progenitor, then we can approximate the number of differentiation-active HSCs with the precursor numbers of LT-HSC.

Second, when Busch et al, 2015 calculated the number of differentiation-active HSC, they measured the cumulative activity of stem cells by following the mice up to 36 weeks postinduction. Our method measured the recent but not accumulative activity of HSC, thus the number of differentiation-active HSC in Busch et al 2015 is predicted to be higher.

Third, Busch et al, 2015 used Tie2MCM Cre to trace HSC. It has been shown that Tie2+ HSC have a higher reconstitution capacity (Ito et al 2016, Science), but no one has compared the *in situ* activity of Tie2+ and Tie2- HSC in a native environment. Since the behavior of HSCs *in situ* may be very different from their behavior in a transplantation setting, it is possible that Tie2+ HSC are more prone to differentiation than Tie2- HSC in a native environment, leading to an overestimation of differentiation-active HSC in the HSC pool.

- Similarly, in Figure 3E, the estimated number of precursors is highest in MPP4, a population typically associated with lymphoid potential and transient myeloid potential, whereas the numbers of MPP3, traditionally associated with myeloid potential, tend to be higher but are not significantly different than those found in HSCs.

We believe this question results from similar confusion of the nomenclature of myeloid precursors in the previous question. As explained previously, the precursors quantified reflect a variety of possible differentiation routes, not just myelopoiesis. Thus, Figure 3E did not suggest that the lymphoid-biased MPP4 has more myeloid precursors than LTHSC. Instead, it simply means more precursors contribute to MPP4 population than the LT-HSC pool. We apologize for the confusion.

- The requirement for estimating precursor numbers at stable levels of Confetti labeling is not well explained. As a result, it is unclear how accurate the estimates of B cell precursors upon transplantation of Fancc-/- cells are. In previous experiments on normal Confetti mice (Figure 3B), the authors do not estimate precursors of lymphopoiesis because Confetti labeling of B cells is not saturated, and this appears to be the case in Fanc-/- animals as well (Fig. 5B).

We appreciate the request for clarification. Our approach required the labeling level to be stable in peripheral blood because we calculate the total number of precursors by normalizing precursor numbers in Confetti+ population with the labeling level (precursor numbers in Confetti+ population divided by labeling efficiency). If the labeling level is not saturated, then the calculation of total precursors will be overestimated. This requirement is more important in native hematopoiesis, since it takes a long time for the mature population, especially the lymphoid population, to be fully replaced by the progenies from the labeled HSPC population (as suggested by Busch et al 2015 and Säwen et al 2018). In transplantation, since lethal irradiation was performed, mature blood cells were rapidly generated by HSPCs, thus saturation of labeling level is not a major concern for precursor quantification. We plan to add Author response image 2 as evidence that Confetti labeling level was stable in mice transplanted with *Fancc-/-* cells.

**Author response image 2. sa4fig2:** 

- Do 9-month-old Fanc-/- animals have reduced lymphoid precursors as well?

Because of the non-saturated labeling in peripheral blood B cells and extra-HSPC induction of Confetti in T cells, we cannot accurately measure lymphoid precursor numbers in 9-month-old *Fancc-/-* animals. As an alternative, the precursor number of lymphoid biased MPP4 population were comparable between *Fancc+/+* and *Fancc-/-* animals (Figure 5D). We plan to add the frequency of common lymphoid progenitors (defined by Lin-IL-7Ra+Sca-1midcKitmid) add a supplementary figure to show were CLP frequencies between these two genotypes.

**Author response image 3. sa4fig3:** 

**Reviewer #2 (Public Review):**
Summary:This manuscript by Liu et al. uses Confetti labeling of hematopoietic stem and progenitor cells in situ to infer the clonal dynamics of adult hematopoiesis. The authors apply a new mathematical framework to analyze the data, allowing them to increase the range of applicability of this tool up to tens of thousands of precursors. With this tool, they (1) provide evidence for the large polyclonality of adult hematopoiesis, (2) offer insights on the expansion dynamics in the fetal liver stage, (3) assess the clonal dynamics in a Fanconi anemia model (Fancc), which has engraftment defects during transplantation.Strengths:The manuscript is well written, with beautiful and clear figures, and both methods and mathematical models are clear and easy to understand.Since 2017, Mikel Ganuza and Shannon McKinney-Freeman have been using these Confetti approaches that rely on calculating the variance across independent biological replicates as a way to infer clonal dynamics. This is a powerful tool and it is a pleasure to see it being implemented in more labs around the world. One of the cool novelties of the current manuscript is using a mathematical model (based on a binomial distribution) to avoid directly regressing the Confetti labeling variance with the number of clones (which only has linearity for a small range of clone numbers). As a result, this current manuscript of Liu et al. methodologically extends the usability of the Confetti approach, allowing them more precise and robust quantification.They then use this model to revisit some questions from various Ganuza et al. papers, validating most of their conclusions. The application to the clonal dynamics of hematopoiesis in a model of Fanconi anemia (Fancc mice) is very much another novel aspect, and shows the surprising result that clonal dynamics are remarkably similar to the wild-type (in spite of the defect that these Fancc HSCs have during engraftment).Overall, the manuscript succeeds at what it proposes to do, stretching out the possibilities of this Confetti model, which I believe will be useful for the entire community of stem cell biologists, and possibly make these assays available to other stem cell regenerating systems.Weaknesses:My main concern with this work is the choice of CreER driver line, which then relates to some of the conclusions made. Scl-CreER succeeds at being as homogenous as possible in labeling HSC/MPPs... however it is clear that it also labels a subcompartment of HSC clones that become dominant with time... This is seen as the percentage of Confettirecombined cells never ceases to increase during the 9-month chase of labeled cells, suggesting that non-labeled cells are being replaced by labeled cells. The reason why this is important is that then one cannot really make conclusions about the clonal dynamics of the unlabeled cells (e.g. for estimating the total number of clones, etc.).

We appreciate the reviewers’ comments. We also agree that this is especially a concern for measuring B cell precursors in native hematopoiesis. For myeloid cells, the increase was much less profound (0.5% per month) after month four post-induction. One way to investigate the dynamics of unlabeled cells is to induce different groups of mice with different doses of tamoxifen so that labeling efficiency varies among different groups. With 14 days of tamoxifen treatment, maximum 60% of HSPC can be labeled (RFP+CFP+YFP). If the unlabeled cells behave similarly with labeled cells, then varying the labeling efficiency shouldn’t affect the total number of precursors calculated (if excluding the potential effect of longer tamoxifen treatment on HSC). While we haven’t extensively performed such lengthy experiment, we have performed one measurement (5 mice) with 14-days of tamoxifen treatment and showed that peripheral blood myeloid precursor numbers calculated from this experiment were comparable to the ones from Figure 3 (2-day tamoxifen).

**Author response image 4. sa4fig4:** 

It's possible that those HSPC that are never labeled with Confetti even during longer tamoxifen treatment could behave differently. In this case, a different Cre driver may provide insight into the total precursor numbers.

I am not sure about the claims that the data shows little precursor expansion from E11 to E14. First, these experiments are done with fewer than 5 replicates, and thus they have much higher error, which is particularly concerning for distinguishing differences of such a small number of clones. Second, the authors do see a ~0.5-1 log difference between E11 and E14 (when looking at months 2-3). When looking at months 5+, there is already a clear decline in the total number of clones in both adult-labeled and embryonic-labeled, so these time points are not as good for estimating the embryonic expansion. In any case, the number of precursors at E11 (which in the end defines the degree of expansion) is always overestimated (and thus, the expansion underestimated) due to the effects of lingering tamoxifen after injection (which continues to cause Confetti allele recombination as stem cell divide). Thus, I think these results are still compatible with expansion in the fetal liver (the degree of which still remains uncertain to me).

We agreed adding additional replicates will reducing any error and boost confidence in our conclusions. The dilemma of comparing fetal- and adult-labeled cohorts is that HSPC activities could not be synchronized among different developmental stages. At fetal to neonatal stage, HSPC proliferate faster to generate new blood cells and support developmental need, while at adult stage HSPC proliferate much slower. Thus, it takes long time for the mature myeloid cells in the adult-labeled cohort to reach a stable Confetti labeling and provide an accurate quantification of precursor. While we agree that it might be better to compare precursor numbers in earlier months, we preferred to compare precursor numbers at later time points for the aforementioned reasons. The other option is to compare the number of HSPC precursors in the BM at earlier time points, as no equilibration of labeling level is required in HSPC, but this requires earlier sacrifice, compromising long term assessment.

We did not revisit questions about the lingering effect of tamoxifen, as this has been studied by Ganuza et al 2017. They showed that tamoxifen was not able to induce additional Confetti recombination if given one day ahead, suggesting the effective window for tamoxifen is less than 24h.

Based on our data, the expansion of lifelong precursors range anywhere from 1.4 to 7.0 (Figure 4G). It’s possible that we might observe a higher level of expansion if the comparison was done in earlier time points. Nonetheless, the assertion that the expansion of life-long HSPC is not as profound as evidenced by transplantation, emphasizes value of HSPC activity analysis *in situ*.

**Reviewer #3 (Public Review):**
Summary:Liu et al. focus on a mathematical method to quantify active hematopoietic precursors in mice using Confetti reporter mice combined with Cre-lox technology. The paper explores the hematopoietic dynamics in various scenarios, including homeostasis, myeloablation with 5-fluorouracil, Fanconi anemia (FA), and post-transplant environments. The key findings and strengths of the paper include (1) precursor quantification: The study develops a method based on the binomial distribution of fluorescent protein expression to estimate precursor numbers. This method is validated across a wide dynamic range, proving more reliable than previous approaches that suffered from limited range and high variance outside this range; (2) dynamic response analysis: The paper examines how hematopoietic precursors respond to myeloablation and transplantation; (3) application in disease models: The method is applied to the FA mouse model, revealing that these mice maintain normal precursor numbers under steady-state conditions and posttransplantation, which challenges some assumptions about FA pathology. Despite the normal precursor count, a diminished repopulation capability suggests other factors at play, possibly related to cell proliferation or other cellular dysfunctions. In addition, the FA mouse model showed a reduction in active lymphoid precursors post-transplantation, contributing to decreased repopulation capacity as the mice aged. The authors are aware of the limitation of the assumption of uniform expansion. The paper assumes a uniform expansion from active precursor to progenies for quantifying precursor numbers. This assumption may not hold in all biological scenarios, especially in disease states where hematopoietic dynamics can be significantly altered. If non-uniformity is high, this could affect the accuracy of the quantification. Overall, the study underscores the importance of precise quantification of hematopoietic precursors in understanding both normal and pathological states in hematopoiesis, presenting a robust tool that could significantly enhance research in hematopoietic disorders and therapy development. The following concerns should be addressed.Major Points:• The authors have shown a wide range of seeded cells (1 to 1e5) (Figure 1D) that follow the linear binomial rule. As the standard deviation converges eventually with more seeded cells, the authors need to address this limitation by seeding the number of cells at which the assumption fails.

While number range above 105 is not required for our measurement of hematopoietic precursors in mice, we agree that it will be valuable to understand the upper limit of experimental measurement. we plan to seed 106-107 cells per replicate to address reviewer’s comments.

• Line 276: This suggests myelopoiesis is preferred when very few precursors are available after irradiation-mediated injury. Did the authors see more myeloid progenitors at 1 month post-transplantation with low precursor number? The authors need to show this data in a supplement.

While we appreciate the concern, we did not generate this dataset because this requires take down of a substantial number of animals at one-month post-transplantation.

Minor Points:• Please cite a reference for line 40: a rare case where a single HSPC clone supports hematopoiesis.• Line 262-263: "This discrepancy may reflect uneven seeding of precursors to the BM throughout the body after transplantation and the fact that we only sampled a part of the BM (femur, tibia, and pelvis)." Consider citing this paper (https://doi.org/10.1016/j.cell.2023.09.019) that explores the HSPCs migration across different bones.• Lines 299 and 304. Misspellings of RFP.

We appreciate reviewer’s suggestions and will modify as suggested.

• The title is misleading as the paper's main focus is the precursor number estimator using the binomial nature of fluorescent tagging. Using a single-copy cassette of Confetti mice cannot be used to measure clonality.

We appreciate reviewer’s suggestions and plan to modify the title of the manuscript to read: “Dynamic Tracking of Native Precursors in Adult Mice”.